# Genome plasticity in *Candida albicans* is driven by long repeat sequences

Robert T Todd[1], Tyler D Wikoff[1], Anja Forche[2], Anna Selmecki[1]*

[1]Creighton University Medical School, Omaha, United States; [2]Bowdoin College, Brunswick, United States

**Abstract** Genome rearrangements resulting in copy number variation (CNV) and loss of heterozygosity (LOH) are frequently observed during the somatic evolution of cancer and promote rapid adaptation of fungi to novel environments. In the human fungal pathogen *Candida albicans*, CNV and LOH confer increased virulence and antifungal drug resistance, yet the mechanisms driving these rearrangements are not completely understood. Here, we unveil an extensive array of long repeat sequences (65–6499 bp) that are associated with CNV, LOH, and chromosomal inversions. Many of these long repeat sequences are uncharacterized and encompass one or more coding sequences that are actively transcribed. Repeats associated with genome rearrangements are predominantly inverted and separated by up to ~1.6 Mb, an extraordinary distance for homology-based DNA repair/recombination in yeast. These repeat sequences are a significant source of genome plasticity across diverse strain backgrounds including clinical, environmental, and experimentally evolved isolates, and represent previously uncharacterized variation in the reference genome.
DOI: https://doi.org/10.7554/eLife.45954.001

*For correspondence:
annaselmecki@creighton.edu

Competing interests: The authors declare that no competing interests exist.

## Introduction

Genome plasticity is surprisingly common in eukaryotes. DNA insertions and deletions (indels), copy number variations (CNV), and loss of heterozygosity (LOH) are frequently described during the evolution of organisms and of disease states such as cancer. In particular, the genome plasticity of fungal pathogens was recognized well before whole genome sequencing was available, including genome copy number variation (polyploidy), inter- and intra-chromosomal rearrangements, and aneuploidy (*Chibana et al., 2000*; *Magee and Magee, 2000*; *Rustchenko-Bulgac, 1991*; *Suzuki et al., 1982*). Controlled in vitro and in vivo evolution experiments in combination with whole genome sequencing have further highlighted the speed in which specific genome rearrangements provide a fitness advantage that can be selected for in these fungal pathogens (*Araya et al., 2010*; *Croll et al., 2013*; *Dunham et al., 2002*; *Forche et al., 2011*; *Ford et al., 2015*; *Gerstein et al., 2015*; *Hirakawa et al., 2015*; *Selmecki et al., 2009*; *Stukenbrock et al., 2010*).

*Candida albicans* is the most prevalent human fungal pathogen, associated with nearly half a million life-threatening infections annually, predominantly in immunocompromised individuals (*Brown and Netea, 2012*). *C. albicans* is a heterozygous diploid yeast capable of mating, yet true meiosis has not been observed. Instead, it undergoes a parasexual process that involves random chromosome loss and rare Spo11-dependent chromosome recombination events (*Bennett and Johnson, 2003*; *Forche et al., 2008*; *Wang et al., 2018*).

The majority of genomic diversity observed in *C. albicans* is attributed to asexual mitotic genome rearrangements (*Forche et al., 2011*; *Lephart and Magee, 2006*). Despite this clonal lifestyle, *C. albicans* isolates exhibit extensive genomic diversity in the form of de novo base substitutions, indels, ploidy variation (haploid, diploid, and polyploid), karyotypic variation due to segmental and whole chromosome aneuploidies, and allele copy number variation including LOH (*Chibana et al.,*

*2000*; *Forche et al., 2011*; *Ford et al., 2015*; *Hickman et al., 2013*; *Hirakawa et al., 2015*; *Magee and Magee, 2000*; *Rustchenko-Bulgac, 1991*; *Selmecki et al., 2006*; *Suzuki et al., 1982*). Additionally, while *C. albicans* did not undergo an ancient whole genome duplication event like *Saccharomyces cerevisiae* (*Butler et al., 2009*; *Marcet-Houben et al., 2009*; *Wolfe and Shields, 1997*), small-scale duplication events have resulted in gene family expansions, especially in sub-telomeric regions (*Anderson et al., 2012*; *Butler et al., 2009*; *Dunn et al., 2018*). A comprehensive analysis of these duplication events, their evolutionary trajectories and impact on genome stability, remains largely unexplored.

Early comparative studies of the *C. albicans* genome identified diverse repetitive loci that contribute to genotypic and phenotypic plasticity (*Braun et al., 2005*; *Jones et al., 2004*). First, repeat analysis in *C. albicans* has characterized at least three major classes of long repetitive sequences: the 23 bp tandem telomeric repeat units and the 14 member telomere-associated (*TLO)* gene family residing in sub-telomeric regions; the Major Repeat Sequences (MRS) found, at least in part, on every *C. albicans* chromosome and formed by a long tandem array of ~2.1 kb RPS units flanking non-repetitive HOK and RBP-2 elements (*Chibana et al., 1994*; *Chindamporn et al., 1998*; *Lephart and Magee, 2006*); and the ribosomal DNA repeats (rDNA) found on ChrR, which are organized as a tandem array of up to ~200 copies of ~12 kb units (*Freire-Benéitez et al., 2016*; *Jones et al., 2004*; *Rustchenko et al., 1993*; *Wickes et al., 1991*). These long repetitive sequences can undergo both inter- and intra-locus recombination events that rapidly generate chromosome length polymorphisms, chimeric chromosomes, and telomere-telomere chromosomal fusions (*Chu et al., 1992*; *Selmecki et al., 2006*; *Selmecki et al., 2010*). Second, like most eukaryotes, *C. albicans* also encodes many 'lone' long terminal repeats (LTRs) and retroelements (Zorro, Tca2, Ty1/Copia) (*Goodwin and Poulter, 1998*; *Goodwin and Poulter, 2000*); however, the relative copy number of many of these genes is hypervariable between *C. albicans* isolates and is expanded relative to other Candida species (*Butler et al., 2009*; *Hirakawa et al., 2015*). Third, short repeat sequences (short tandem repeats and trinucleotide repeats) are significantly more frequent in protein-coding sequences of *C. albicans* than in *S. cerevisiae* and *Schizosaccharomyces pombe* (*Braun et al., 2005*; *Jones et al., 2004*). Fourth, expansions of multi-gene families (identified by protein alignment) were both more common and larger than the orthologous gene family size found in *S. cerevisiae*. These gene families often encode proteins with roles in commensalism and virulence, including the agglutinin-like sequence (*ALS*) family (eight genes) and other glycosylphosphatidylinositol (GPI)-anchored genes that encode large cell surface glycoproteins (five genes) (*Levdansky et al., 2008*; *Wilkins et al., 2018*). Among these gene families, recombination and/or slippage between repeat units yields extensive allelic variation, leading to functional and phenotypic diversity, similar to the *FLO* genes in *S. cerevisiae* (*Hoyer et al., 1995*; *Kunkel, 1993*; *Pearson et al., 2005*; *Richard et al., 1999*; *Verstrepen et al., 2005*; *Zhang et al., 2003*; *Zhao et al., 2004*). The evolution of different alleles in these repeat-containing ORFs predominantly occurs by the addition, deletion, and rearrangement of repeat units within an ORF and between different ORFs, not by the acquisition of point mutations or indels (*Christiaens et al., 2012*; *Zhang et al., 2010*). Importantly, these genomic studies focused on short repeat sequences and repeats found in protein-coding sequences. Less is known about long repeat sequences found throughout the genome, especially those encoding multiple ORFs and intergenic regions.

Over 19 years ago, Wolfe and colleagues showed that the *C. albicans* genome contains thousands of small chromosomal inversion events (~10 genes long) relative to *S. cerevisiae*. These inversions resulted in substantially different gene order between these two species (*Seoighe et al., 2000*). Similarly, Dujon and colleagues demonstrated that the *C. albicans* genome had the highest rate of genome instability due to micro- and macro-rearrangements of syntenic gene blocks, relative to 11 other hemiascomycete species (*Fischer et al., 2006*). The loss of synteny primarily resulted from chromosomal rearrangements, not sequence divergence of orthologous regions. A mechanism proposed for this genome instability was a higher incidence of repetitive sequences and/or a less efficient DNA repair process (*Fischer et al., 2006*).

The genomic diversity of *C. albicans* increases during in vitro and in vivo exposure to stress. For example, rates of LOH increase during exposure to elevated temperature (37°C), DNA transformation, and antifungal drugs (*Bouchonville et al., 2009*; *Forche et al., 2011*; *Forche et al., 2018*). LOH is also increased during in vivo models of infection (*Ene et al., 2018*; *Forche et al., 2008*; *Forche et al., 2018*). LOH events occur due to chromosome nondisjunction leading to whole

chromosome LOH or via recombination, in which only part of the chromosome undergoes LOH. Exposure to stress also selects for isolates that have acquired adaptive mutations and genome rearrangements. For example, aneuploidy is found in ~50% of isolates resistant to the most common antifungal drug, fluconazole (FLC). The most common and only recurrent aneuploidy in different strain backgrounds is the amplification of the left arm of chromosome 5 (Chr5L), often through acquisition of a novel isochromosome structure (denoted as i(5L)), comprised of two copies of Chr5L separated by the centromere (*Selmecki et al., 2006*; *Selmecki et al., 2008*). Acquisition of i(5L) conferred FLC resistance via the amplification of two genes, *ERG11* and *TAC1*, encoding the drug target (Erg11) and a transcriptional activator of drug efflux pumps (Tac1) (*Selmecki et al., 2008*; *Selmecki et al., 2009*). Importantly, the centromere of Chr5 contains a long inverted repeat sequence, and recombination between these repeats can form homozygous isochromosomes of both the left arm (i(5L)) and right arm of Chr5 (i(5R)) (*Selmecki et al., 2006*). The role of long repeat sequences in the formation of other segmental aneuploidies and other genome rearrangements has not been comprehensively addressed.

We provide evidence that long repeat sequences are involved in the formation of all observed CNV breakpoints and chromosome inversions, and many LOH breakpoints, across 33 diverse clinical and experimentally evolved isolates. Our comprehensive analysis of long repeat sequences within the *C. albicans* genome identified hundreds of sequences representing novel multicopy repeats, none of which include MRS, rDNA, sub-telomeric repeats, known repeat families (*ALS*, *TLOs*) or known repetitive elements (tRNAs, LTRs, retrotransposons). Long repeats that are associated with genome rearrangements (CNV, LOH, and inversions) have on average higher sequence identity than all long repeats combined. Additionally, long repeats that contain ORFs (including partial ORF sequences, single complete ORF sequences (paralogs), or multiple ORFs and intergenic sequences) are longer and associated with more genome rearrangements than long repeats that contain other genomic features (such as LTRs, retrotransposons, or tRNAs). Additionally, repeat copies involved in genome rearrangements can be located up to ~1.6 Mb apart on the same chromosome, suggesting a non-conventional, long-range mechanism for DNA double-strand break (DSB) repair and somatic genome diversification.

## Results

### An inverted repeat within *CEN4* is associated with the formation of a novel isochromosome

To identify the mechanisms by which *C. albicans* isolates generate genome plasticity, we performed a comparative genomics analysis of 33 diverse clinical isolates (*Supplementary file 1*). This set of isolates included 11 that underwent controlled experimental evolution, where a known progenitor isolate was passaged in vitro or in vivo. Additionally, we performed comparative genomics on newly obtained clinical isolates, and clinical isolates whose genomes were published previously, including the reference isolate SC5314.

Given the significant impact of i(5L) on antifungal drug resistance, we focused first on the characterization of a novel segmental aneuploidy detected on Chr4 that arose during in vitro evolution in the presence of FLC. Initially, we passaged a FLC-sensitive clinical isolate P78042, which was trisomic for Chr4 (*Hirakawa et al., 2015*; *Lockhart et al., 2002*), in the presence of FLC (128 μg/ml) for 100 generations by serial dilution (see Materials and methods). One evolved isolate (AMS3743) was selected, based on increased fitness in FLC (see below), and the whole genome was sequenced. Read depth analysis indicated that this isolate had four copies of the right arm of Chr4 (Chr4R), but only two copies of Chr4L, and the copy number breakpoint occurred at the centromere of Chr4 (*CEN4*) (*Figure 1A*). Wildtype *CEN4*, like *CEN5*, is comprised of a CENP-A-binding core sequence (~3.1 kb) flanked by a long (524 bp) inverted repeat (*Burrack et al., 2016*; *Ketel et al., 2009*; *Sanyal et al., 2004*).

To test the hypothesis that this segmental aneuploidy is an isochromosome structure, we performed CHEF karyotype analysis. Isolate AMS3743 had a novel ~1.2 Mb chromosome band that hybridized to a *CEN4* probe via Southern blot (*Figure 1B*). This ~1.2 Mb band was twice the size of the right arm of Chr4 (~607 Kb). Consistent with an isochromosome i(4R) structure (a centromere flanked by inverted copies of Chr4R), a single primer amplified a ~4.1 kb product, from Chr4R

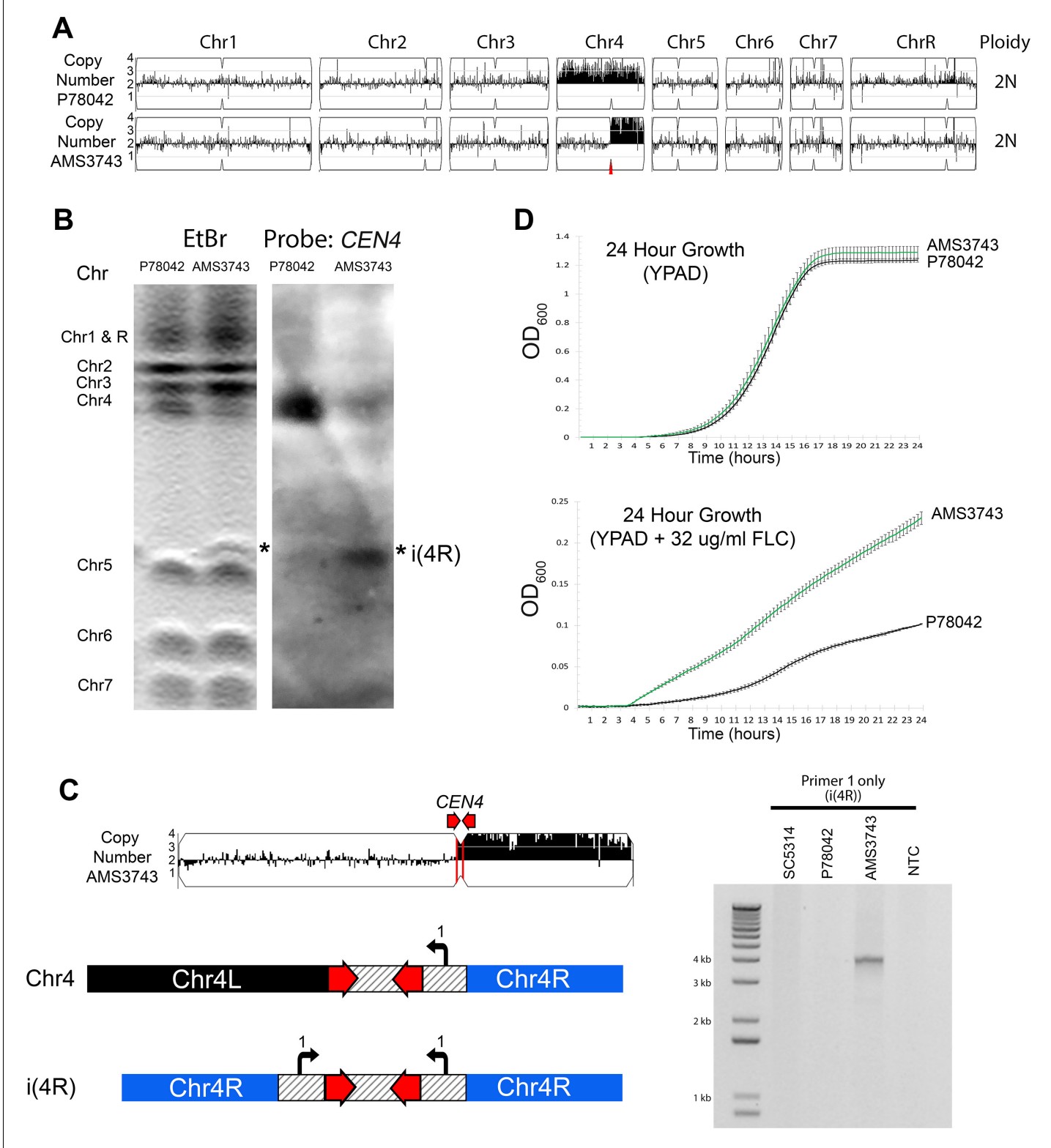

**Figure 1.** Inverted repeat at *CEN4* causes a novel isochromosome leading to increased fluconazole resistance. (**A**) Whole genome sequence data plotted as a log2 ratio and converted to chromosome copy number (Y-axis) and chromosome location (X-axis) using YMAP, for the progenitor clinical isolate (P78042) and an isolate obtained after 100 generations in FLC (AMS3743). The copy number breakpoint in AMS3743 occurs at *CEN4* (red arrow). (**B**) CHEF karyotype gel stained with ethidium bromide (left panel) identifies a novel band (asterisk) above Chr5. Southern blot analysis (right panel) of the same gel using a DIG-labeled *CEN4* probe identifies the full-length Chr4 homolog in P78042 and AMS3743, and the novel band in AMS3743 that is

*Figure 1 continued on next page*

*Figure 1 continued*

twice the size of the right arm of Chr4 in an isochromosome structure (asterisk, i(4R)). (**C**) PCR validation of i(4R). Schematic representation of the Chr4 homologue (top) and i(4R), where the location of a single primer sequence (Primer 1, ***Supplementary file 7***) that flanks the *CEN4* inverted repeat is indicated. PCR with Primer 1 amplified the expected product of i(4R) in AMS3743. (**D**) 24 hr growth curves in YPAD (top panel) and YPAD +32 µg/ml FLC (bottom panel) for P78042 (black line) and AMS3743 (green line). Average slope and standard error of the mean for three biological replicates is indicated. The average maximum slope (n = 3) of P78042 and AMS3743 in YPAD was not significantly different (0.046 and 0.046, respectively, p>0.75, t-test). The average maximum slope (n = 3) of P78042 and AMS3743 was significantly different in FLC (0.002 and 0.003, respectively, p<0.0006, t-test). OD, optical density (***Figure 1—source data 1***).

DOI: https://doi.org/10.7554/eLife.45954.002

The following source data and figure supplements are available for figure 1:

**Source data 1.** Growth curve analysis.
DOI: https://doi.org/10.7554/eLife.45954.005

**Figure supplement 1.** Long inverted repeats on Chr4 are associated with a centromere inversion and an isochromosome that confers increased fitness in FLC.
DOI: https://doi.org/10.7554/eLife.45954.003

**Figure supplement 2.** Sanger sequencing of *CEN4* in SC5314.
DOI: https://doi.org/10.7554/eLife.45954.004

through *CEN4* and back to Chr4R in the isolate with i(4R) but did not amplify any sequence in the reference (SC5314), or progenitor (P78042) isolates (***Figure 1C***).

Next, we determined the impact of i(4R) on fitness in the presence and absence of FLC over a 24 hr period. In the presence of FLC, the i(4R) isolate grew significantly better than the progenitor P78042 (p<0.0006, t-test, ***Figure 1D***). Interestingly, in the absence of FLC, the i(4R) isolate grew as well as the progenitor P78042 (***Figure 1D***). Furthermore, i(4R) was maintained in 12/12 populations for over ~300 generations in the absence of FLC (see Materials and methods). One of the populations, AMS3743_10, appeared to be losing i(4R) as measured by CHEF gel densitometry (see Materials and methods) and was plated for single colonies in the absence of FLC. One colony (out of six) had lost i(4R) (AMS3743_10_S6, ***Figure 1—figure supplement 1A***). To ask if i(4R) was necessary and sufficient for the increased fitness in FLC, fitness was determined in the presence and absence of FLC. The colony that had lost i(4R) had a reduced growth rate in the presence of FLC, similar to the progenitor P78042 (***Figure 1—figure supplement 1B***).

Overall, these data imply that the long inverted repeat within *CEN4* can generate an independent isochromosome structure comprised of two right arms of Chr4, and that i(4R) is necessary and sufficient for increased fitness in FLC. These results parallel the identification of isochromosomes associated with the long inverted repeat sequence within *CEN5,* which can result in the formation of i(5R) and i(5L), the latter of which confers FLC resistance (***Selmecki et al., 2006***; ***Selmecki et al., 2008***).

## Inverted repeat sequences are associated with inversion of centromere sequences

During our investigation of the i(4R) structure, we unveiled a surprising feature of *CEN4*: the CENP-A-binding core sequence of *CEN4* contained two different alleles. One homologue of Chr4 contained a ~3.1 kb sequence inversion between the inverted repeat associated with *CEN4*. The new, inverted *CEN4* sequence was detected by PCR in the reference isolate SC5314, and in the distantly related isolates P78042 and AMS3743 (***Figure 1—figure supplement 1C & D***). Sanger sequencing indicated that a recombination event occurred between the two arms of the inverted repeat (***Figure 1—figure supplement 2***). Interestingly, the CENP-A-binding core sequence of *CEN4* is asymmetrically positioned on one side of the inverted repeat sequence (***Figure 1—figure supplement 1D***, shaded region) (***Burrack et al., 2016***; ***Sanyal et al., 2004***). Therefore, this inversion caused a separation between the known CENP-A-binding core sequence of *CEN4* that is located to the right and outside of the inverted repeat.

## Identification of long repeat sequences throughout the *C. albicans* genome

Given the extensive genome rearrangements observed at the long inverted repeat associated with *CEN4*, we sought to characterize all long repeat sequences within the *C. albicans* reference

genome (SC5314). All long sequence matches within SC5314 were identified by aligning the reference genome sequence to itself using the bioinformatics suite MUMmer (*Kurtz et al., 2004*). First, all exact sequence matches of 20 nucleotides or longer were identified, then all matches were clustered and extended to obtain a maximum-length colinear string of matches, resulting in a final list of long repeat sequences that ranged from 65 bp to 6499 bp (median 318 bp) with sequence identities of $\geq$80% (See Materials and methods). The genomic position and percent identity of all matched repeats was determined with MUMmer and manually verified using BLASTN and IGV (*Robinson et al., 2011*; *Thorvaldsdóttir et al., 2013*). After excluding all rDNA, MRS and sub-telomeric repeat sequences, 1974 long repeat matches were identified (*Supplementary file 2*). The MUMmer analysis identified five ORFs and one gene family with known, complex embedded tandem repeat sequences (*PGA18*, *PGA55, EAP1, orf19.1725, CSA1,* and the *ALS* gene family, herein referred to as 'the complex tandem repeat genes'). The complexity of these repeat sequences prohibited the assignment of exact repeat copy number per genome, and they were removed from analyses when indicated. The remaining long repeat sequences cover 2.87% of the haploid reference genome (see Materials and methods).

Long repeat matches occurred between sequences on the same chromosome (intra-chromosomal repeats, *Figure 2A*), on different chromosomes (inter-chromosomal repeats), or both. The number of all repeat matches per chromosome was correlated with chromosome size ($R^2$ = 0.65, p<0.016, *Figure 2B*); however, regions of high repeat density (e.g. ChrRR near the rDNA) or low repeat density (e.g. Chr7L) were detected on some chromosome arms. This repeat density did not correlate with GC content ($R^2$ = 0.063, p>0.32) or ORF density ($R^2$ = 0.02, p>0.59) on any chromosome arm (*Figure 2—source data 1*).

We next calculated the orientation and distance between matched intra-chromosomal repeat sequences (*Figure 2—figure supplement 1*), both important factors for reconstructing the evolutionary history of these duplication events and for analyzing the frequency and outcome of homologous recombination events that occur between repeat sequences (*Lobachev et al., 1998*; *Ramakrishnan et al., 2018*). Intra-chromosomal repeats are often generated in tandem by recombination between sister chromatids or replication slippage, and these repeats can move further away from each other by chromosomal rearrangement events (including chromosomal inversions) (*Achaz et al., 2000*; *Reams and Roth, 2015*). Indeed, intra-chromosomal repeats were predominantly tandem, although inverted and mirrored repeats also occurred (*Supplementary file 2*). We hypothesized that the distance between matched intra-chromosomal repeats (spacer length) would be predominantly short and that the distribution of spacer lengths on each chromosome would be similar. Strikingly, spacer length ranged from 1 bp to 2,856,212 bp (median ~82.8 kb, excluding the complex tandem repeat genes, see Materials and methods), and was correlated with chromosome size (*Figure 2—figure supplement 2A*, $R^2$ = 0.066, p<0.0001). Additionally, the distribution of spacer lengths was significantly different between chromosomes (*Figure 2—figure supplement 2B*, p<0.035, Kruskal-Wallis with Dunn's multiple comparison test) with the larger chromosomes (Chr1 and ChrR) containing many repeat matches that were separated by distances greater than ~1.5 Mb. The increased distance between repeat sequences likely occurred via additional large inversions, insertions or telomere-telomere recombination/fusion events.

We further annotated the long repeat sequences according to the genomic features contained within each repeat (see Materials and methods). The most common long repeats contained lone long terminal repeats (LTRs) (775), followed by ORFs (339, excluding the complex tandem repeat genes), tRNAs (334), and retrotransposons (40). Repeat matches containing ORFs included partial ORF sequences (196/339, 57.8%), single complete ORF sequences (114/339, 33.6%), and multiple ORFs and intergenic sequences (29/339, 8.6%) (*Supplementary file 2*). Repeat matches containing complete ORFs and multiple ORFs represent paralogs and multi-gene duplication events. Additionally, there were 349 intergenic, unannotated sequences, 231 that shared high-sequence identity (>83%) with an annotated sequence found elsewhere in the genome, including known LTRs, retrotransposons, and ORFs (*Supplementary file 2*, 'Unannotated Intergenic Sequence'). For example, an additional 54 LTRs were identified in the reference genome with this analysis. Interestingly, LTR matched repeat pairs were predominantly dispersed on different chromosomes (78%), while ORF matched repeat pairs were predominantly located on a single chromosome (64%, *Figure 2C*).

Of the matched repeat pairs, the long repeat sequences containing ORFs had the lowest median sequence identity when compared to repeats containing other features (*Figure 2—figure*

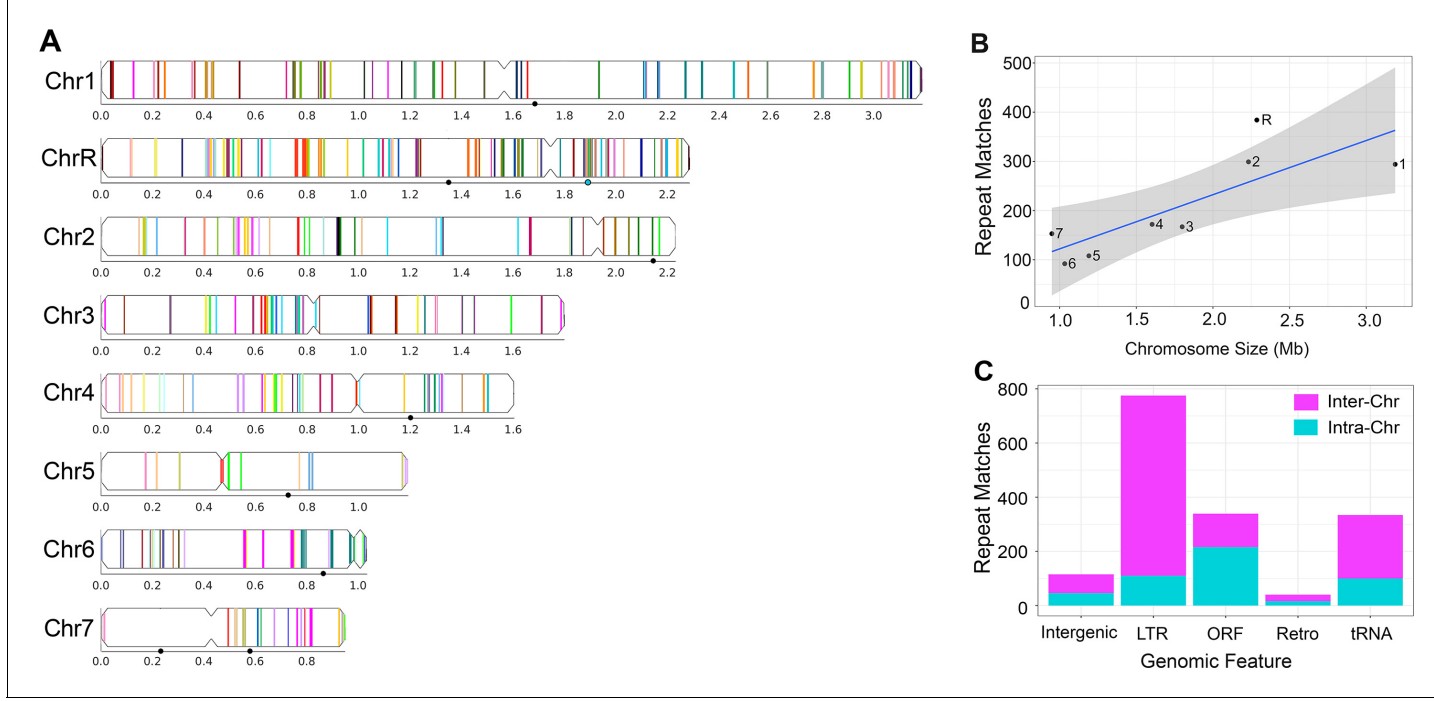

**Figure 2.** Long repeat sequences are found across the *C. albicans* genome. Detailed results for all long intra- and inter-chromosomal repeat positions, orientations, and gene features are found in *Supplementary file 2*. Repeats associated with the rDNA, major repeat sequences (MRS), and sub-telomeric repeats were removed prior to the analysis. (**A**) Representative image of the long intra-chromosomal repeat positions (colored lines – not to scale). Each repeat family is assigned a unique color within its respective chromosome. Numbers and symbols below each chromosome indicate chromosomal position (Mb), MRS position (black circles), and rDNA locus (blue circle, ChrR). (**B**) Number of all repeat matches (excluding the complex tandem repeat genes) on each chromosome, ordered by chromosome size ($R^2 = 0.65$, p<0.016, gray indicates 95% confidence interval, *Figure 2—source data 1*). (**C**) The number of intra-chromosomal (Intra-Chr) and inter-chromosomal (Inter-Chr) repeat matches assigned to each genomic feature: Intergenic, LTR, ORF (excluding the complex tandem repeat genes), retrotransposon (Retro), and tRNA (*Figure 2—source data 1*).
DOI: https://doi.org/10.7554/eLife.45954.006

The following source data and figure supplements are available for figure 2:

**Source data 1.** Distribution, features, and coverage of long repeat sequences in *C. albicans*.
DOI: https://doi.org/10.7554/eLife.45954.010
**Source data 2.** Analysis of long repeat spacer length in *C. albicans*.
DOI: https://doi.org/10.7554/eLife.45954.011
**Source data 3.** Analysis of key features of long repeat sequences in *C. albicans*.
DOI: https://doi.org/10.7554/eLife.45954.012
**Figure supplement 1.** Features of long repeat sequences.
DOI: https://doi.org/10.7554/eLife.45954.007
**Figure supplement 2.** The intra-chromosomal repeats with the longest spacer length are found on the longer chromosomes.
DOI: https://doi.org/10.7554/eLife.45954.008
**Figure supplement 3.** Key features of long repeat sequences in *C. albicans*.
DOI: https://doi.org/10.7554/eLife.45954.009

*supplement 3A*, p<0.0001, Kruskal-Wallis with Dunn's multiple comparison test). Conversely, repeats containing ORFs had significantly longer copy length than any other genomic feature (p<0.0001, Kruskal-Wallis with Dunn's multiple comparison test) and was the only feature that had a significant increase in copy length of intra-chromosomal matches relative to inter-chromosomal matches (*Figure 2—figure supplement 3B*, p<0.0001, Kruskal-Wallis with Dunn's multiple comparison test). The long repeat sequences containing ORFs were predominantly present in only two copies per genome, had pairwise coding sequences with similarly high identity, and therefore represent paralogous gene duplication events (*Supplementary file 2*). The origin, function, and evolutionary trajectory of these paralogs may provide insight into the evolution of fungal pathogens like *C.*

*albicans* that did not undergo the ancient whole genome duplication event (*Butler et al., 2009*; *Marcet-Houben et al., 2009*; *Wolfe and Shields, 1997*).

The complex tandem repeat genes, for which genome copy number could not be determined, had low sequence identity and were predominantly found on Chr6 (*Figure 2—figure supplement 3C*). In contrast, the full-length coding sequence of all ORFs that were contained within long repeat sequences, were significantly longer (median value of 1380 bp vs 1200 bp, *Figure 2—figure supplement 3D*, p<0.0008, Kolmogorov-Smirnov test) and had a significantly higher GC content (median value of 37.22% vs 35.22% *Figure 2—figure supplement 3E*, p<0.0001, Kolmogorov-Smirnov test) than the full-length coding sequence of all ORFs not contained within long repeat sequences (genome-wide, excluding the complex tandem repeat genes, see Materials and methods). Interestingly, increased GC content was correlated with increased rates of both mitotic and meiotic recombination events in *S. cerevisiae* (*Kiktev et al., 2018*).

## Identification of CNV breakpoints in isolates with segmental aneuploidies

Next, CNV breakpoints were determined across 13 additional isolates with one or more segmental aneuploidies. Six of these isolates were from in vitro evolution experiments in the presence of azole antifungal drugs (FLC or miconazole), four were from in vivo evolution experiments in a murine model of oropharyngeal candidiasis (OPC) performed in the absence of antifungal drugs, and three were human clinical isolates (*Supplementary file 1*). All segmental aneuploidies arose from a known euploid diploid progenitor (*Abbey et al., 2014*; *Hirakawa et al., 2015*), except two clinical isolates with unknown origin and the i(4R) isolate that arose from a trisomic progenitor, described above.

Segmental aneuploidies were initially detected by CHEF karyotype analysis and ddRAD-seq, but the coordinates of the CNV breakpoints were not known (*Abbey et al., 2014*; *Forche et al., 2018*; *Mount et al., 2018*; *Ropars et al., 2018*). The ploidy of each isolate was measured by flow cytometry and the DNA copy number of all loci was determined using whole genome sequencing (see Materials and methods). Among the 13 diverse isolates, 19 segmental aneuploidies were confirmed, with at least one segmental aneuploidy detected on each of the eight chromosomes (*Figure 3A*, *Figure 3—figure supplement 1A–J*). Segmental amplifications were more frequent (12/19, 63.2%) than segmental deletions (3/19, 15.8%). The remaining segmental aneuploidies (4/19, 21.1%) consisted of more complex rearrangements that resulted in a segmental amplification and a terminal chromosome deletion at the same breakpoint.

## All segmental aneuploidies occur at long repeat sequences

The CNV breakpoint of each segmental aneuploidy was determined using both read depth and allele ratio analysis (see Materials and methods). From the 19 segmental aneuploidies, 26 CNV breakpoints were identified because some segmental aneuploidies contained multiple breakpoints. Strikingly, every CNV breakpoint occurred within 2 kb of a long repeat sequence, ranging from 248 bp to ~4.76 kb in length. Observed breakpoints had significantly more overlap with long repeat sequences than expected given the total genome coverage of long repeat sequences (p<0.0001, two-tailed Fishers Exact Test, see Materials and methods). All but one of the repeat sequences were intra-chromosomal and separated by a distance ranging from ~3.1 kb to ~1.62 Mb (*Supplementary file 3*). Importantly, repeats containing ORFs were significantly more common than all other types of repeats at these breakpoints (18/26 CNV breakpoints, p<0.001, $\chi^2$ Goodness-of-fit test).

Three examples of CNV breakpoints in long repeats containing ORFs were observed in isolates AMS3053, AMS3420 and CEC2871. In both AMS3053 and AMS3420, a long inverted repeat sequence was associated with a complex segmental amplification and a terminal chromosome deletion that resulted in a long-range homozygosis event. In AMS3053, the breakpoint on Chr3L occurred within a ~1.7 kb inverted repeat sequence (>99% identity) separated by ~11.5 kb (*Figure 3B*). The left side of this inverted repeat contained four uncharacterized ORFs (*orf19.279*, *orf19.280*, *orf19.281*, *orf19.284*) and associated intergenic sequences, while the right side contained three uncharacterized ORFs (*orf19.296*, *orf19.295*, *orf19.292*) and one characterized ORF (*orf19.297 DTD2*) plus associated intergenic sequences. Similarly, the OPC-derived isolate AMS3420 underwent a complex segmental amplification and deletion within a ~1.6 kb inverted repeat sequence on Chr1L

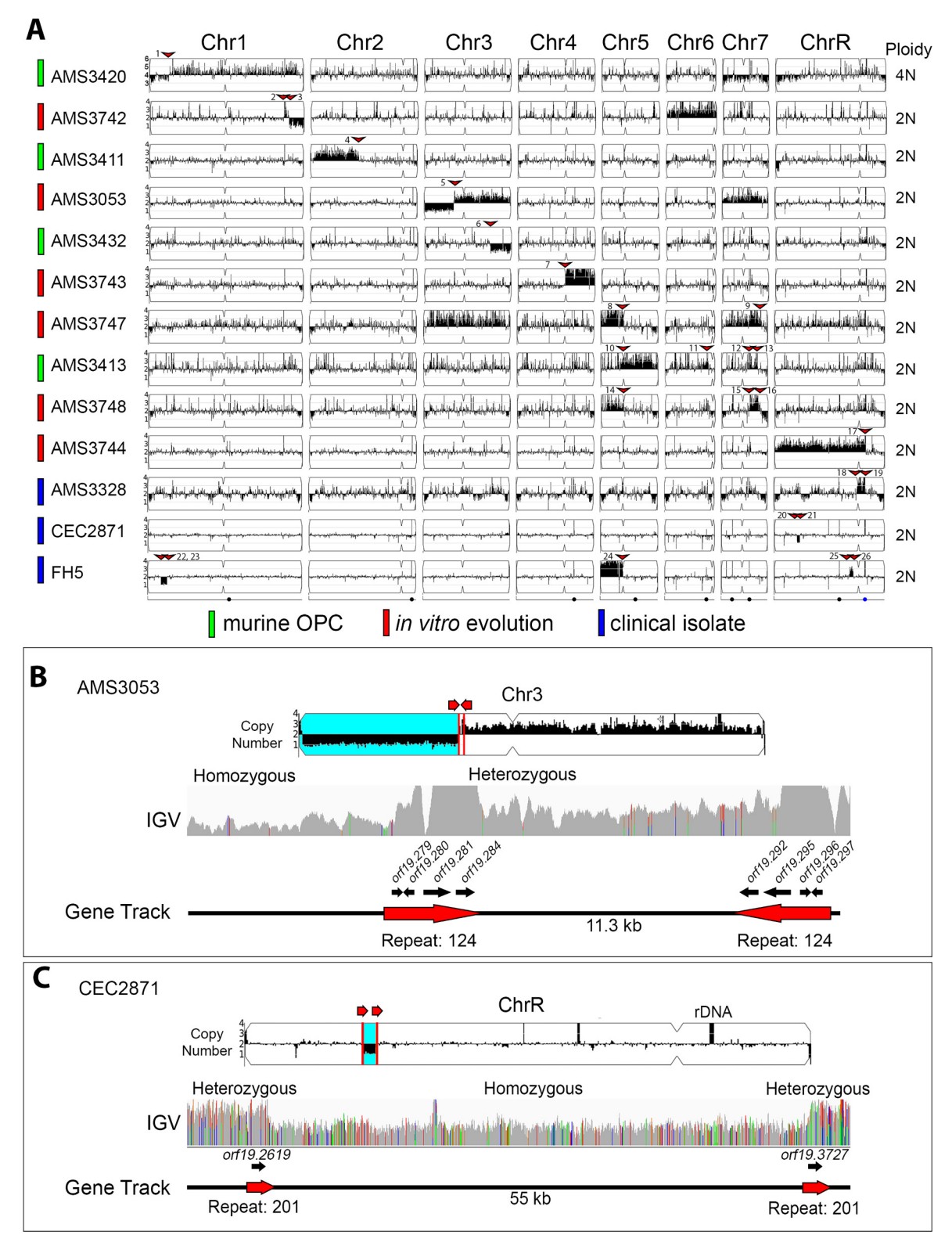

**Figure 3.** All copy number breakpoints resulting in segmental aneuploidy occur at repeat sequences. (**A**) Whole genome sequence data plotted as a log2 ratio and converted to chromosome copy number (Y-axis) and chromosome location (X-axis) using YMAP. The source of each isolate is indicated in color: in vivo evolution experiments in a murine model of oropharyngeal candidiasis (OPC) (green), in vitro evolution experiments in the presence of azole antifungal drugs (red), and clinical isolates (blue). Ploidy, determined by flow cytometry, is indicated on the far right. Every copy number

*Figure 3 continued on next page*

*Figure 3 continued*

breakpoint occurred at a repeat sequence (red arrow), additional details are in *Supplementary file 3*. Location of the Major Repeat Sequences (black circle) and rDNA array (blue circle) are shown below. Example copy number breakpoints for two isolates (B–C). (B) Isolate AMS3053 underwent a complex rearrangement on Chr3L at a long inverted repeat (Repeat 124, red lines). Read depth (top panel) and allele frequency (IGV panel) data indicate the copy number breakpoint coincided with LOH (blue region) telomere proximal to the breakpoint. The inverted repeat copies (~3.2 kb, 99.5% sequence identity, separated by ~11.3 kb) each contain four complete ORFs and intergenic sequences. (C) Read depth (top panel) and allele frequency (IGV panel) data for isolate CEC2871 shows an internal chromosome deletion on ChrR with copy number breakpoints (red lines) and LOH (blue) that occur between a long tandem repeat (Repeat family 201, red arrows). The tandem repeat copies (~1.4 kb, 93.8% sequence identity, separated by ~55 kb) each contain one ORF.
DOI: https://doi.org/10.7554/eLife.45954.013

The following figure supplement is available for figure 3:

**Figure supplement 1.** Segmental aneuploidies occur at previously characterized and uncharacterized long repeat sequences.
DOI: https://doi.org/10.7554/eLife.45954.014

(91.5% identity) separated by ~26 kb, which contains the high-affinity glucose transporters *HGT1* and *HGT2* (*Figure 3—figure supplement 1A*). Long internal chromosome deletions were also observed. For example, in isolate CEC2871, a ~55 kb deletion resulted from recombination between a ~1.4 kb tandem repeat on ChrR (92.4% identity) containing ORFs of the *PHO* gene family (*PHO112* and *PHO113*, *Figure 3C*). Proposed models for recombination events that would result in these complex segmental amplifications and deletions are described in the discussion.

Eight CNV breakpoints occurred within other long repeat sequences, including: a ~200 bp microsatellite repeat (1/26), intergenic repeats (1/26), MRS (2/26), LTRs (2/26), and the rDNA repeats (2/26) (*Figure 3A*, *Supplementary file 3*). Some segmental aneuploidies were comprised of multiple breakpoints, each associated with a different repeat family (e.g. *Figure 3—figure supplement 1I & J*). Interestingly, both breakpoints that occurred at the rDNA also amplified the ChrR centromere (*CENR*), and everything either to the telomere of the opposite chromosome arm (ChrRL) (*Figure 3—figure supplement 1H*), or to a microsatellite repeat sequence on ChrRL (AMS3328, *Figure 3A*).

In summary, all CNV breakpoints in this collection occurred at or within long repeat sequences. Inverted repeat sequences predominantly coincided with segmental amplifications and terminal chromosome deletions, while tandem repeat sequences coincided with internal chromosome deletions. Some aneuploidies were comprised of multiple breakpoints, each associated with a different repeat family. Overall, a repeat homology-associated repair mechanism appears to be driving the formation of segmental aneuploidies. Importantly, the involvement of long repeats in CNV breakpoints is independent of genetic background and environmental selection.

## LOH occurs at long inter- and intra-chromosomal repeat sequences

In many of the isolates with segmental aneuploidies, the CNV also was accompanied by LOH (e.g. *Figure 3B and C*). To ask if long repeat sequences were associated with LOH breakpoints in the absence of detectable CNVs, we selected 20 near-euploid genomes that had at least one long-range homozygous region, but the coordinates of the LOH breakpoint were not known (*Ford et al., 2015*; *Hirakawa et al., 2015*; *Ropars et al., 2018*). These 20 isolates belong to nine major *C. albicans* clades from different origins (e.g. superficial and invasive human infections, healthy human hosts, and spoiled food) (*Figure 4A*, *Supplementary file 1*).

153 LOH breakpoints were identified in the 20 isolates (See Materials and methods, *Supplementary file 4*). 61/153 LOH breakpoints were found within 2 kb of a long repeat sequence, and, like the CNV breakpoints, these LOH breakpoints could occur on any chromosome (*Figure 4A*). The copy length of the repeat sequences found at LOH breakpoints ranged from 78 bp to 6499 bp (median 516 bp) with sequence identities ranging from 82.2% to 100% (median of 95.1%). Most of the repeats associated with LOH breakpoints were intra-chromosomal (46/61), in all three orientations (inverted, mirrored, and tandem), and separated by a distance ranging from 903 bp to ~1.6 Mb (median ~35.3 kb). The vast majority of long-range homozygous regions contained only one LOH breakpoint and proceeded from the breakpoint to the proximal telomere, similar to previous analyses (*Ene et al., 2018*; *Forche et al., 2008*; *Forche et al., 2009*; *Selmecki et al., 2005*). Surprisingly, four isolates had an LOH breakpoint that proceeded from one chromosome arm to the

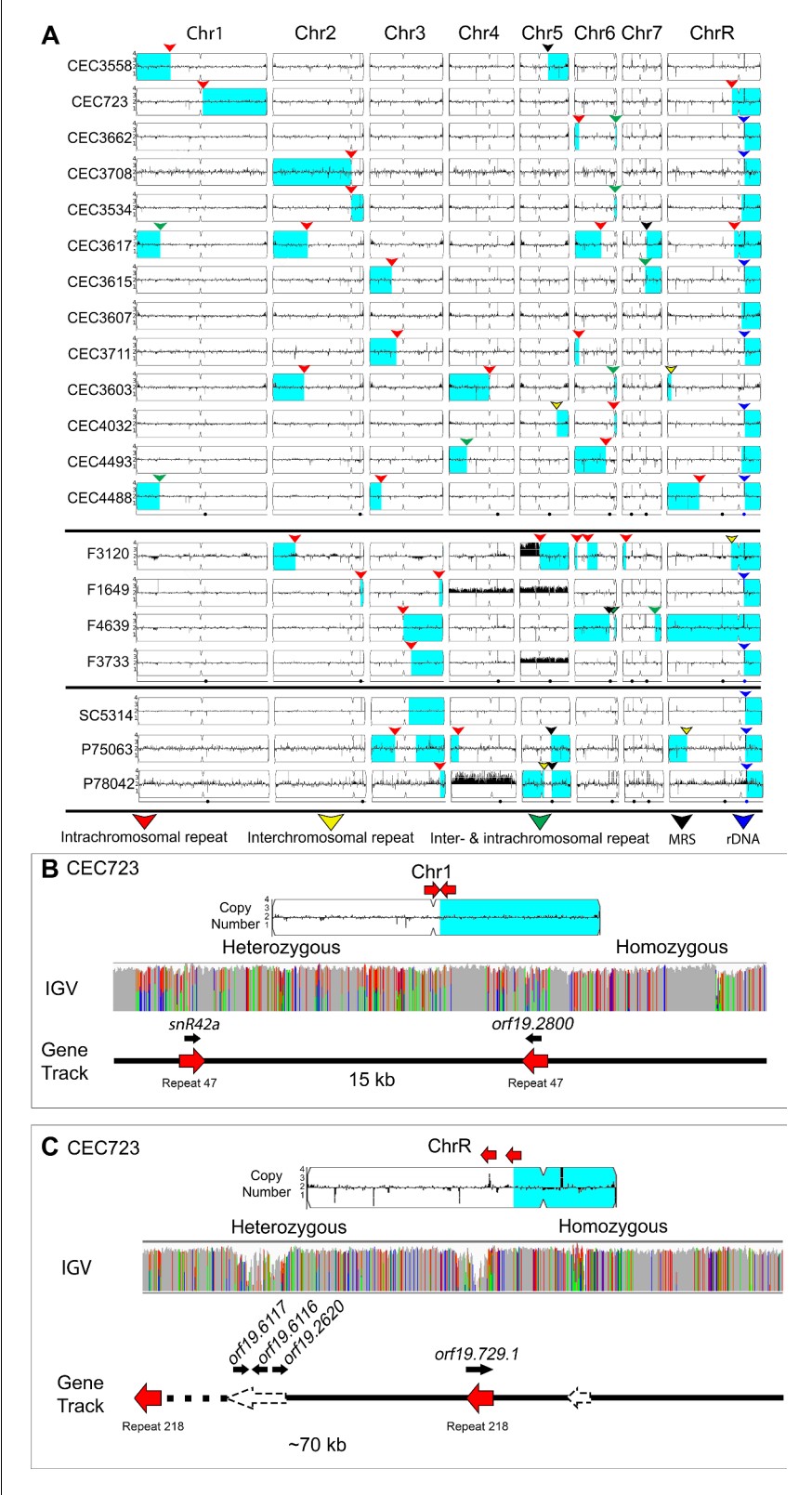

**Figure 4.** Many LOH breakpoints occur at long intra- and inter-chromosomal repeat sequences. Whole genome sequence data plotted as a log2 ratio and converted to chromosome copy number (Y-axis) and chromosome location (X-axis) using YMAP. (**A**) All long-range homozygous regions (light blue) that are associated with long repeat sequences (colored arrows) are indicated for 20 diverse *C. albicans* isolates. LOH breakpoints and isolate information are detailed in *Supplementary files 1* and *4*. The type of long repeat is indicated with colored arrows: intra-chromosomal (red), inter-
*Figure 4 continued on next page*

*Figure 4 continued*

chromosomal (yellow), both intra- and inter-chromosomal (green), rDNA repeat (blue), and MRS (black). (B–C) Two example LOH breakpoints in isolate CEC723 that occur at long repeats (red arrows) on (B) Chr1 (repeat copy length ~1.1 kb), and (C) ChrR (repeat copy length ~3.3 kb) and continue to the right telomere of the respective chromosomes. Heterozygous and homozygous allele ratios are indicated in the IGV track. The position, orientation, and spacer length of the long repeat sequence is indicated in the gene track. ORFs (black arrows) contained within the long repeat sequences are indicated above the gene track. The LOH breakpoint on ChrR is within a repeat-dense region; additional long repeats in the region are indicated (dashed arrows).

DOI: https://doi.org/10.7554/eLife.45954.015

The following figure supplement is available for figure 4:

**Figure supplement 1.** Long-track homozygosis occurs on Chr3L at telomere seed sequences.

DOI: https://doi.org/10.7554/eLife.45954.016

telomere on the opposite chromosome arm, causing centromere homozygosis (three events on ChrR and one event on Chr5).

One isolate, CEC723, had two long-range homozygous regions associated with intra-chromosomal repeat sequences. The first LOH breakpoint on Chr1R was associated with a ~1.1 kb mirrored repeat sequence (>99% identity) separated by ~15 kb (*Figure 4B*). One copy of the repeat sequence contained a snoRNA (*snR42a*) and the other contained an uncharacterized ORF (*orf19.2800*), which we predict also encodes a second copy of *snR42a*. The second LOH breakpoint on ChrRL was associated with a ~3.2 kb tandem repeat sequence (97.7% identity) separated by ~70 kb (*Figure 4C*). This breakpoint was flanked by additional long repeat sequences that were associated with CNV in other isolates, indicating that this region is a hotspot for genome rearrangements (*Supplementary file 2*).

Finally, the reference isolate SC5314 contains a well-known long-range homozygous region on Chr3R. We asked if this LOH breakpoint occurred within a long repeat sequence. Remarkably, the LOH breakpoint occurred in *orf19.5880* near an 8 bp sequence (AACTTCTT) identical to part of the *C. albicans* 23 bp telomere repeat sequence (GGTGTACGGATTGTCT<u>AACTTCTT</u>). Furthermore, a second copy of this same 8 bp sequence was found in an inverted orientation ~3.4 kb away in the adjacent ORF (*orf19.5884*). This long-range LOH event continued to the right telomere of Chr3. While LOH may have resulted from a repair template on the other homolog, an alternative model cannot be ruled out. We previously found that an LOH and CNV breakpoint that caused a segmental Chr5 truncation in the common laboratory strain BWP17 (*Selmecki et al., 2005*) was initiated at a 9 bp sequence (CTAACTTCT) that is almost identical to the sequence found at this breakpoint (AACTTCTT). We posit that a similar chromosome truncation, followed by reduplication of the monosomic portion of Chr3 (*Figure 4—figure supplement 1A & B*) may have generated the homozygosis of Chr3. These 8 bp and 9 bp telomere-like sequences occur 2160 and 249 times, respectively, within the non-telomeric portions of the *C. albicans* reference genome (*Supplementary file 5*). The presence of such a large number of potential template sequences, especially if including the telomere repeats at each chromosome end, might have driven this two-step model.

## Repeat sequences cause sequence inversions and heterozygous islands

As expected, levels of heterozygosity were high within long repeat sequences due to the ability of short-read (Illumina) sequences to map to multiple positions in the genome (e.g. the heterozygous bases within repeat sequences in *Figure 4B and C*). Unexpectedly, between or adjacent to some long repeat sequences, heterozygous islands were observed in otherwise homozygous regions of the genome. For example, in isolate P75063, an LOH breakpoint on Chr4L was associated with a ~1.7 kb inverted repeat and resulted in a terminal homozygosis of the chromosome (*Figure 5A*). Adjacent to this homozygous region was an ~32 kb region that had multiple homozygous/heterozygous transitions (5′ homozygous-heterozygous-homozygous-heterozygous 3′). We hypothesized that a long sequence inversion, similar to that observed within the repeats flanking *CEN4*, accounted for the multiple heterozygous to homozygous transitions in this region. PCR amplification between unique sequences flanking the inverted repeat revealed a ~32 kb inversion in P75063 and SC5314 and was the only orientation that amplified by PCR; the reference orientation did not amplify, suggesting that the reference genome may be incorrect at this position (*Figure 5B*).

These two long inversions (at *CEN4* and Chr4L), plus an additional seven potential sequence inversions were identified bioinformatically from a set of 21 clinical isolates (*Hirakawa et al., 2015*);

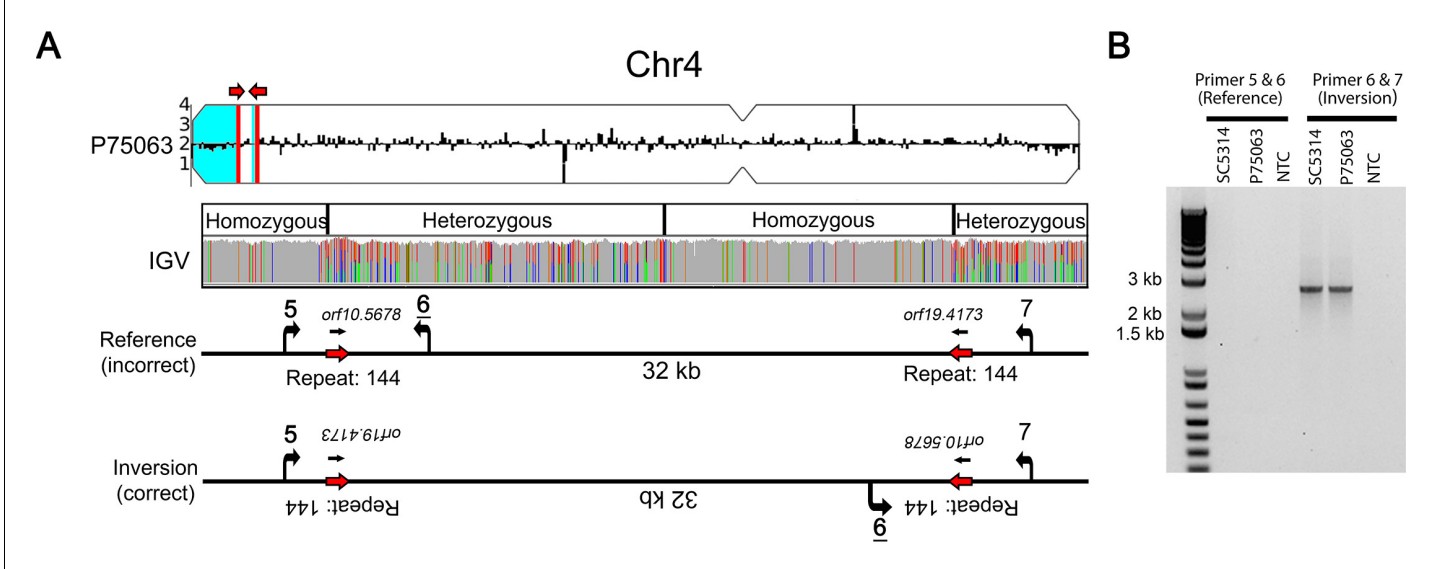

**Figure 5.** Long repeat sequences are associated with chromosomal inversions. (**A**) Whole genome sequence read depth plotted as a log2 ratio and converted to chromosome copy number (Y-axis) and chromosome location (X-axis) using YMAP. Long-range homozygous regions (blue) on Chr4 are indicated for the isolate P75063. IGV allele ratio track indicates multiple homozygous to heterozygous transitions between a long inverted repeat (red arrows, repeat 144, copy length ~1.7 kb). Primers (5, 6, and 7, **Supplementary file 7**) were designed to unique sequences flanking repeat 144. (**B**) PCR amplification between Primers 6 and 7 identifies a ~32 kb chromosomal inversion in both the reference isolate SC5314 and P75063; the reference orientation did not amplify (Primers 5 and 6).

DOI: https://doi.org/10.7554/eLife.45954.017

however, none of these inversion breakpoints were characterized or validated by PCR or Sanger sequencing. We found that all potential inversions had breakpoints within long inverted repeats, and these potentially cause chromosomal inversions of ~4.1 kb to ~102.6 kb in length (median ~39.0 kb, **Supplementary file 6**). All but one sequence inversion (8/9) occurred within repeats containing ORFs and a high median sequence identity (98.3%). In summary, we identified examples of chromosomal inversions that occurred between long repeat sequences and provide the first molecular validation of these inversions in both the reference SC5314 and clinical isolates.

## Breakpoints resulting in CNV, LOH, and inversion, occur in the longest repeat sequences with highest homology

Overall, many uncharacterized long repeat sequences exist within the *C. albicans* genome. Repeats associated with breakpoints (CNV, LOH, and inversion) were significantly longer than all other long repeat sequences (median copy length of 785 bp vs 278 bp, p<0.0001, Kolmogorov-Smirnov test), and had a significantly higher percent sequence identity than all other long repeat sequences (median identity of 96.2% vs 94.2%, p<0.036 Kolmogorov-Smirnov test) (**Figure 6A**). Repeats containing ORFs were longer than repeats containing other genomic features and were the most common repeat identified at breakpoints (33/53, 62.3%, **Figure 6B & C**). Furthermore, repeats containing ORFs were the only genomic feature with both significantly longer copy length and significantly higher sequence identity at breakpoints than at non-breakpoints (p<0.0001 copy length, p<0.0001 sequence identity Kolmogorov-Smirnov test, **Figure 6—figure supplement 1A & B**). Additionally, repeat matches that contain multiple ORF sequences represent only 8.6% of all long repeats containing ORFs, yet these extra-long repeats comprise 26.8% of the observed breakpoints (**Supplementary file 2**). Therefore, at least under selection, genome rearrangements are occurring more often at repeats with high sequence identity, and at repeats with high sequence identity and high copy length, the latter of which includes ORFs.

Nine repeat families were associated with more than one breakpoint type (CNV, LOH, and inversion), and two of these (124 and 151) were associated with all three breakpoint types. Repeat family 124 (**Figure 3B and 6A**), comprised of 4 ORFs, was one of the longest repeats (~3.2 kb) and had

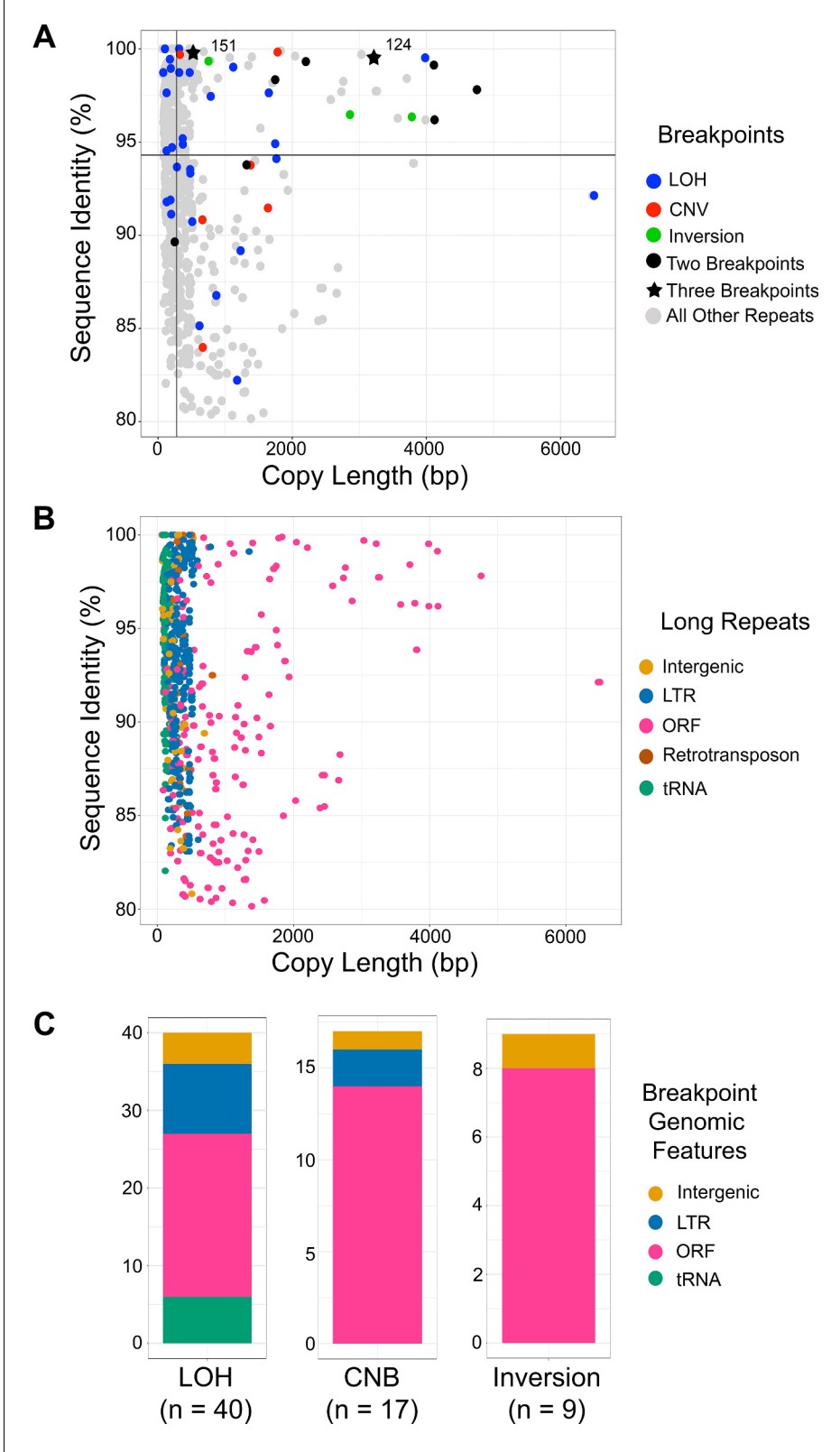

**Figure 6.** Breakpoints associated with CNV, LOH, and inversion predominantly occur at long repeats that contain ORFs. (**A**) Scatterplot of percent sequence identity and copy length of all long repeat matches in *Supplementary file 2*, excluding the complex tandem repeat genes. All long repeats are indicated in gray, and repeats associated with the observed breakpoints are indicated as follows: LOH (blue), CNV (red), and inversion (green). Six repeats (black circle) were associated with more than one type of breakpoint, and two repeats (black star) were associated with all three types of

*Figure 6 continued on next page*

*Figure 6 continued*

breakpoints. Solid black lines indicate the median repeat copy length (278 bp, vertical black line) and median percent sequence identity (94.3%, horizontal black line). Repeats associated with LOH, CNV, and inversion breakpoints have a significantly higher median copy length (p<0.0001, Kolmogorov-Smirnov test) and median sequence identity (p<0.036, Kolmogorov-Smirnov test) than all other long repeat sequences (excluding the complex tandem repeat genes, *Figure 6—source data 1*). (B) Scatterplot as in *Figure 6A*, where genomic features contained within long repeats are indicated: intergenic sequence (light brown), lone LTR (blue), ORF (pink), retrotransposon (dark brown), and tRNA (green). (C) The distribution of genomic features contained within long repeats at LOH, CNV, and inversion breakpoints. Colors indicated as in *Figure 6B*.

DOI: https://doi.org/10.7554/eLife.45954.018

The following source data and figure supplement are available for figure 6:

**Source data 1.** Analysis of long repeat sequences associated with CNV, LOH, and sequence inversion.
DOI: https://doi.org/10.7554/eLife.45954.020
**Figure supplement 1.** Breakpoint-associated repeats containing ORFs have both high sequence identity and long copy length.
DOI: https://doi.org/10.7554/eLife.45954.019

one of the highest percent sequence identities (>99%). Repeat family 151 flanks *CEN4* and was associated with the formation of the novel isochromosome i(4R), which was necessary and sufficient for increased fitness in the presence of FLC (*Figure 1C* and *Figure 6A*). Overwhelmingly, these data support that long repeat sequences found throughout the *C. albicans* genome are utilized to generate segmental aneuploidies, long-range LOH and sequence inversions, and that in at least one environment these rearrangements provide a significant fitness benefit to the organism.

## Discussion

Genomic variation caused by CNV, LOH, and sequence inversion can drive rapid adaptation and promote tumorigenesis. Here, we examined the role of genome architecture during the formation of genetic variation in the diploid, heterozygous fungal pathogen, *C. albicans*. Our genome-wide analysis of 33 isolates identified long repeat sequences that had prominent roles in generating genomic diversity. These long repeats included previously uncharacterized repeat sequences, centromeric repeats, repeats found within intergenic sequences, and repeats that span multiple ORFs and intergenic sequences. Importantly, long repeat sequences were found at every CNV and sequence inversion breakpoint observed, and frequently occurred at LOH breakpoints as well. Long repeats that were associated with all breakpoints (CNV, LOH, and inversion) had on average significantly higher sequence identity compared to all repeats identified (p<0.036, Kolmogorov-Smirnov test). Furthermore, repeats containing ORFs had both significantly higher sequence identity and significantly longer copy length at breakpoints than at non-breakpoints (sequence identity p<0.0001, copy length p<0.0001 Kolmogorov-Smirnov test, *Figure 6*, *Figure 6—figure supplement 1A and B*). These results were independent of genetic background or source of isolation. Thus, long repeat sequences found across the *C. albicans* genome underlie the formation of significant genome variation that can increase fitness and drive adaptation.

### DNA double-strand breaks are repaired using long repeat sequences found across the *C. albicans* genome

The genomic variants described in this study are the result of DNA double-strand breaks (DSBs) and subsequent recombination events resulting in CNVs, LOH, and sequence inversions. While the factors leading to, and the location of the initiating DSBs are unknown, the genomic variants recovered were all selected as viable, and perhaps beneficial, outcomes of the DSB repair process. DSBs are repaired by either non-homologous end-joining (NHEJ) or homologous recombination (HR). HR is thought to be a high-fidelity repair process due to the use of an intact, homologous DNA template. However, recent studies have also implicated HR in an increased rate of mutagenesis and chromosomal rearrangements (*Bishop and Schiestl, 2000*; *Kramara et al., 2018*).

We also found that the orientation of repeat copies had a major effect on the outcome of the genome rearrangements observed. Inverted repeat sequences frequently were found within 2 kb of chromosomal amplification events, while tandem repeat sequences frequently were found within 2 kb of long internal chromosomal deletions. We propose two models of HR involved in the production of genome variation observed in this study (*Figure 7*).

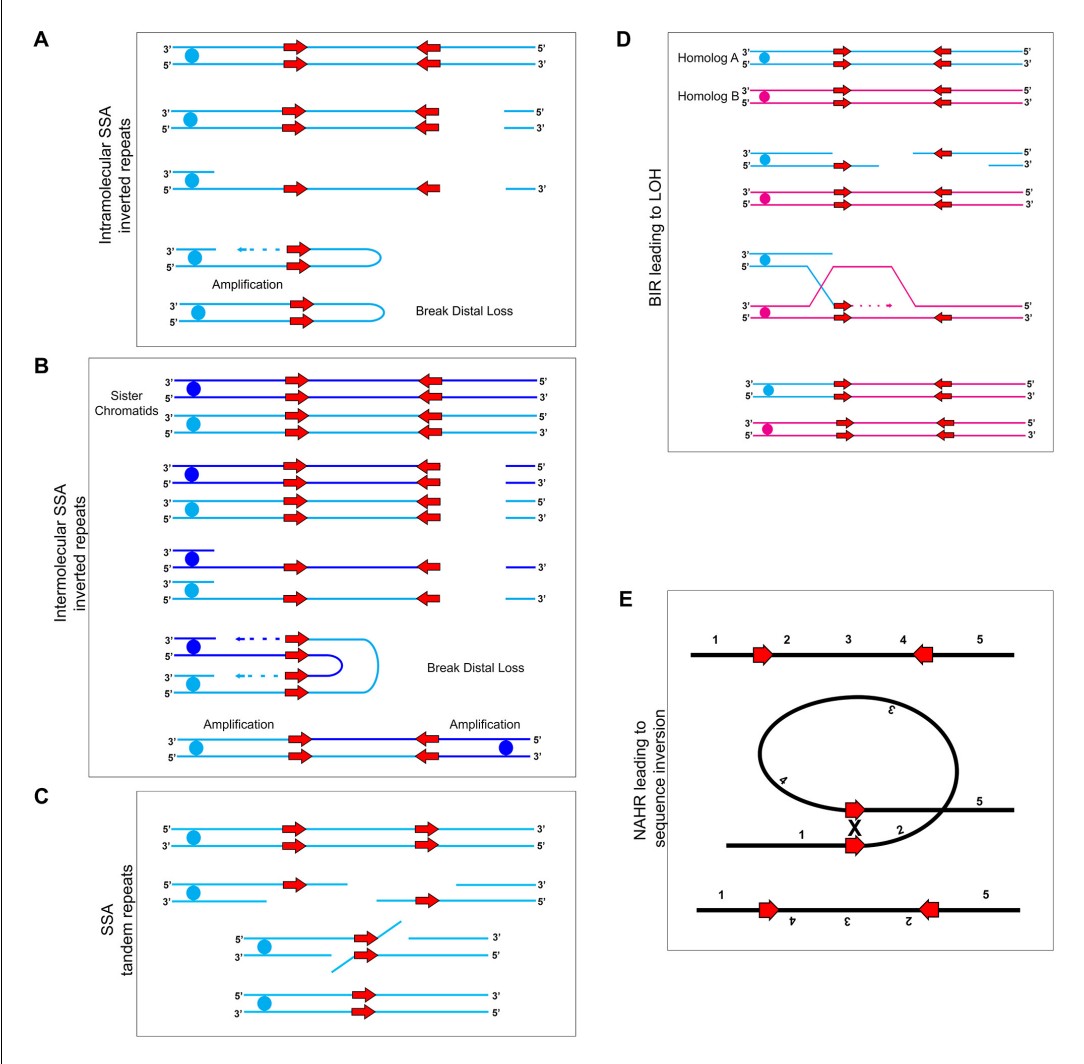

**Figure 7.** Mechanisms for recombination between long repeats that result in segmental amplification, deletion, LOH, and/or inversion. (A) Intra-molecular single-strand annealing occurs after a double strand break (DSB) on a single DNA molecule undergoes 5′—3′ resection exposing two copies of an inverted repeat on the single-stranded 3′ overhang. Annealing of the two inverted repeat copies occurs followed by DNA synthesis resulting in a fold-back structure and partial chromosome truncation. (B) Inter-molecular single-strand annealing occurs when a DSB occurs on two separate DNA molecules. After 5′—3′ resection, annealing between the single-stranded inverted repeat copies of the two different DNA molecules results in the formation of a dicentric chromosome and partial chromosome truncation. (C) A single DNA molecule (blue) containing two tandem repeats (red arrows) undergoes a DSB leading to 5′—3′ resection that exposes the tandem repeats. The homologous sequences anneal and non-homologous 3′ tails are removed. The remaining gap is filled producing an intact chromosome that has undergone an internal deletion. (D) Break-Induced Replication (BIR) induces LOH between repeat sequences found on opposite homologs: Two homologs, homolog A (blue) and homolog B (magenta), contain inverted repeat sequences (red arrows). A DSB occurring on homolog A leads to strand invasion and DNA synthesis. Upon termination of synthesis of both the leading and lagging strands, all sequences to the right of the DSB are homozygous. (E) Inversion events occur due to intra-molecular recombination between inverted repeats (red arrows) flanking a unique sequence. The orientation of the reference sequence is indicated above chromosome (1-2-3-4-5). Non-Allelic Homologous Recombination (NAHR) between the inverted repeats leads to an inversion of the sequence between the repeats (1-4-3-2-5).

DOI: https://doi.org/10.7554/eLife.45954.021

First, we propose that single-strand annealing (SSA) is initiated by the annealing of DNA repeats that become single-stranded after a DSB and 5′—3′ DNA resection (*Figure 7A–7B*) and occurs between both tandem and inverted repeat sequences (*Bhargava et al., 2016*; *Malkova and Haber, 2012*; *Mehta and Haber, 2014*; *Ramakrishnan et al., 2018*; *VanHulle et al., 2007*). SSA that occurs between tandem repeats leads to segmental deletion of the sequence located between the repeat

sequences (*Figure 7C*). SSA that occurs between inverted repeats can lead to the formation of complex, often unstable dicentric and 'fold-back' chromosomes which then enter the breakage-fusion-bridge cycle leading to further genome instability (*Aguilera and García-Muse, 2013*; *Croll et al., 2013*; *McClintock, 1939*; *McClintock, 1941*; *McClintock, 1942*; *VanHulle et al., 2007*) (*Figure 7A–7B*). Evidence for dicentric chromosomes may exist in several isolates that acquired a segmental amplification of the centromere (*Figure 3*); however, we do not know from these data if the amplification is on the same molecule (generating a dicentric chromosome) or elsewhere in the genome.

The second HR mechanism we propose is break-induced replication (BIR) which is initiated by DSBs that have only one free end available for repair. During BIR, single-stranded DNA invades a homologous sequence followed by subsequent DNA synthesis which can copy long, chromosomal-sized DNA segments (*Anand et al., 2013*; *Kramara et al., 2018*; *Malkova and Ira, 2013*; *Mehta and Haber, 2014*). If templating and synthesis occurs on a homologous chromosome, BIR can lead to long-range homozygosis of a chromosome (*Figure 7D*). Processes similar to BIR have been proposed for CNV generation in a diverse set of organisms ranging from bacteria to humans (*Hastings et al., 2009*). These predominantly micro-homology-mediated BIR (MMBIR) events use short regions of homology to repair DSBs in a Rad51-independent manner (*Hastings et al., 2009*). One caveat is that the repeat sequences involved in generating genome rearrangements observed in this study are much longer than those involving MMBIR. While repair by BIR is rare in *S. cerevisiae* model systems, the selective benefit of the resulting genotypes generated by BIR could increase the apparent frequency with which these types of mutations are recovered in certain environments, for instance the acquisition of i(4R) in the presence of FLC (*Figure 1*).

## *C. albicans* repeat copy length and spacer length

The repeat copy length associated with observed breakpoints in *C. albicans* are similar in copy length to transposable (Ty) elements in *S. cerevisiae* (~6 kb) and long interspersed nuclear elements (LINE) in the human genome (~6–7 kb), which are a major source of genome rearrangements (*Chen et al., 2014*; *Dunham et al., 2002*; *Gresham et al., 2010*; *Higashimoto et al., 2013*; *Selmecki et al., 2015*). Both Ty and LINE elements are high copy number repeats; LINE elements are present in thousands of copies in the human genome (*Rodić and Burns, 2013*). However, beyond the similarity in copy length, we rarely found high copy number repeats, like lone LTRs or retrotransposons, associated with CNV and inversion breakpoints (5.7%, *Figure 6*). These breakpoints predominantly occurred at repeats containing ORFs that are often present in only two copies per genome (*Supplementary file 2*). LOH breakpoints, on the other hand, were associated more often with LTRs (22.6%, *Figure 6*), which may be a result of selection or may suggest a preference for a different repair mechanism when a DSB occurs near these loci.

The repeat copy length and spacer length associated with the observed breakpoints in *C. albicans* are much longer than typically observed in *S. cerevisiae*. Segmental amplification events in *S. cerevisiae* are often mediated by short inverted repeat sequences, for example, 8 bp long and separated by 40 bp (*Brewer et al., 2011*; *Lauer et al., 2018*; *Payen et al., 2014*; *Sunshine et al., 2015*). The presence of a short, inverted repeat sequence within a replication fork can stimulate ligation between the leading and lagging strands, which results in replication and formation of an extra-chromosomal circle. This extra-chromosomal amplification may continue to replicate independently if it contains an origin of replication (defined as origin-dependent inverted-repeat amplification [ODIRA]; *Brewer et al., 2015*; *Brewer et al., 2011*; *Payen et al., 2014*). It seems unlikely that such a mechanism operates at the long distances observed between repeat sequences in *C. albicans*. However, it is possible that a different origin-dependent mechanism is mediating some of the rearrangements we observed (see centromere discussion below). A future challenge is to determine if/how this occurs.

The spacer length, especially between inverted repeats, has been a major focus of genome instability research. Identification and characterization of inverted repeats in *S. cerevisiae* has primarily focused on those repeats that are separated by very short (~80 bp) spacers (*Strawbridge et al., 2010*). Inverted repeats that were engineered to have variable repeat spacer lengths identified a correlation between repeat and spacer length and DSB repair. Increasing repeat copy length (from 185 bp to ~1.5 kb) and/or decreasing repeat spacer length (from ~8.5 kb to 0 bp) increased the recombination rate between repeats by up to 17,000-fold (*Lobachev et al., 1998*). Furthermore, spacer length alone could affect the choice of DSB repair pathway; DSB repair via inter-molecular

SSA predominantly occured with a spacer length of 1 kb, while intra-molecular SSA predominantly occured with spacer length of 12 bp (*Ramakrishnan et al., 2018*).

Astoundingly, the *C. albicans* CNV and inversion breakpoints are associated with much longer repeat spacer lengths than those described in *S. cerevisiae*, ranging from ~3.1 kb to ~1.6 Mb (median ~30 kb) and ~3.1 kb to ~94.3 kb (median ~34.6 kb), respectively. Recombination between such long distances requires a naturally occurring, long-distance homology search. It is tempting to speculate that *C. albicans* may have a mechanism for long-distance resection, particular chromatin features, or a 3D-nuclear structure that facilitates recombination between inverted repeats separated by long distances.

## Inverted repeat sequences directly associated with the CENP-A-binding centromere core sequences facilitate isochromosome formation

Centromeres were common breakpoints for CNV, LOH and inversion. Twelve of the 33 isolates had breakpoint events that occurred within centromeres, including those described at *CEN4* and *CEN5*, as well as two additional centromeres that contained one copy of a long repeat sequence, *CEN2* and *CEN3* (*Supplementary file 2*). Notably, *C. albicans* centromeres are the earliest firing centers of DNA replication (*Koren et al., 2010*; *Tsai et al., 2014*). Therefore, errors in DNA replication may be a common source of DSBs that are repaired via HR between long repeat sequences.

Repair of a DSB within or near a centromere-associated inverted repeat can result in isochromosome formation or centromere inversion (*Figure 1*, *Figure 1—figure supplement 1*). Both of the *C. albicans* centromeres that are flanked by long inverted repeat sequences (*CEN4* and *CEN5*) can form isochromosomes (*Figure 1* and *Selmecki et al., 2006*; *Selmecki et al., 2009*). Exposure to the antifungal drug FLC selected for isochromosome formation at both *CEN4* and *CEN5*. If a DSB occurs near the inverted repeat sequence, DNA synthesis via BIR will copy the entire arm of the broken chromosome, resulting in the homozygous isochromosome structures that we observed (*Figure 1* and *Selmecki et al., 2010*; *Selmecki et al., 2009*). Acquisition of either isochromosome i(4R) or i(5L) was both necessary and sufficient for increased fitness in the presence of FLC (*Figure 1* and *Selmecki et al., 2006*). Additionally, there was no fitness cost associated with either isochromosome in the absence of FLC: i(4R) was stable for ~300 generations in 12/12 populations in the absence of FLC (*Figure 1—figure supplement 1*). These data are in contrast to other, often whole chromosome and multiple chromosome aneuploidies that cause significant fitness defects in the absence of selection (*Pavelka et al., 2010*; *Torres et al., 2007*), but support observations that aneuploidy in general has less of a fitness cost in diploid and polyploid fungi (*Hose et al., 2015*; *Scott et al., 2017*; *Selmecki et al., 2015*; *Tan et al., 2013*).

Similarly, repair of a DSB within or near a centromere-associated inverted repeat can result in centromere inversion. Inversions are the result of intra-chromosomal non-allelic homologous recombination between inverted repeats flanking the centromere (*Figure 7E*). Here, we detected an inversion that occurred between inverted repeats flanking *CEN4*. The impact of these inversions on localization of the centromeric histone CENP-A, or of the recombination proteins Rad51 and Rad52, which are thought to recruit CENP-A, are not known. Whether or not inversion of the centromere affects chromosome stability will be important to test in future experiments.

In this study, Illumina short-read datasets were used to identify genomic features that were driving structural and allelic variation across diverse *C. albicans* isolates. The use of both new and previously published short-read datasets highlights the utility of this bioinformatic approach for the analysis of structural variants within this and other species. However, short-read data are unable to provide a key understanding of the molecules containing the long repeat sequences. For example, the definitive structure of chromosomal inversions, including the heterozygous *CEN4* sequence, is difficult to determine with short-read data. PCR enabled rapid validation of these inversions (*Figures 1* and *5*); however, it required knowledge of the repeat location and unique surrounding sequences. Future long-read sequencing is needed to address the definitive structure of existing DNA molecules and potential DNA intermediates involved in recombination and resolution of CNV, LOH, and inversions.

## Long repeats containing ORFs were significantly more common at breakpoints resulting in CNV, LOH and inversion than any other genomic feature

One hypothesis is that active transcription may promote DNA DSBs, due to the formation of R-loop structures (*Aguilera and Gaillard, 2014*; *Santos-Pereira and Aguilera, 2015*). Additionally, increased transcription in certain environments may increase the probability of a DNA DSB that result in genome rearrangements, as was observed at the *S. cerevisiae CUP1* locus in high-copper environments (*Adamo et al., 2012*; *Fogel et al., 1983*; *Hull et al., 2017*; *Thomas and Rothstein, 1989*). Several indirect results are consistent with this hypothesis in *C. albicans*. First, all ORFs within a long repeat that were associated with a breakpoint were indeed actively transcribed in the reference isolate SC5314 during growth in rich medium (*Bruno et al., 2010*). Second, some breakpoint ORFs have increased expression in the selective environment from which the isolate with the breakpoint was obtained. For example, two different in vivo isolates, one bloodstream clinical isolate and one murine OPC-evolved isolate, have the same breakpoint on Chr1 at the inverted repeat that includes *HGT1* and *HGT2* (*Supplementary file 2*). Both *HGT1* and *HGT2* are induced during OPC, biofilm production and adaptation to serum (*Horák, 2013*; *Nobile et al., 2012*; *Pitarch et al., 2001*). Therefore, increased transcription of these repeat ORFs in vivo is a potential source of DNA damage that resulted in DSB repair.

### Conclusion

In conclusion, genome rearrangements resulting in segmental aneuploidies, sequence inversions, and LOH are associated with long repeat sequence breakpoints on every chromosome. These genome rearrangements can arise rapidly, both in vitro and in vivo, and can provide an adaptive phenotype such as improved growth in antifungal drugs. Importantly, long repeat sequences are hotspots for genome variation across diverse selective environments. Indeed, several repeats were involved in all three types of genome rearrangements in different isolates. These data support the idea that the *C. albicans* genome is one of the most rapidly evolving genomes due to disruption of conserved syntenic sequence blocks via genome rearrangements between long repeat sequences (*Fischer et al., 2006*). Finally, given the frequency of long repeat sequences in the human genome, studies of *C. albicans* genome rearrangements can contribute to understanding the mechanisms that facilitate CNV, LOH, and inversions associated with human disease and cancer.

## Materials and methods

**Key resources table**

| Reagent type (species) or resource | Designation | Source or reference | Identifiers | Additional information |
|---|---|---|---|---|
| Strain, strain background (*Candida albicans*) | SC5314 | *Hirakawa et al., 2015* (DOI: 10.1101/gr.174623.114) | RRID:SCR_013437 | |
| Strain, strain background (*C. albicans*) | P78042 | *Hirakawa et al., 2015* (DOI: 10.1101/gr.174623.114) | | |
| Strain, strain background (*C. albicans*) | AMS3743 | This Study | | In vitro evolution of P78042 in 128 ug/ml FLC for 100 generations |
| Strain, strain background (*C. albicans*) | AMS3743_10 | This Study | | In vitro evolution of AMS3743 in rich medium for 300 generations |
| Strain, strain background (*C. albicans*) | AMS3743_10_S6 | This Study | | Single colony from AMS3743_10 |
| Antibody | Anti-Digoxigenin-AP Fab Fragments | Roche | 11093274910 RRID:AB_2734716 | (1:5000) |

*Continued on next page*

Continued

| Reagent type (species) or resource | Designation | Source or reference | Identifiers | Additional information |
|---|---|---|---|---|
| Sequenced-based reagent | PCR Primers | This Study | | *Supplementary file 7* |
| Commercial assay or kit | Illumina Nextera XT Library Prep Kit | Illumina | 105032350 | |
| Commercial assay or kit | Illumina Nextera XT Index Kit | Illumina | 105055294 | |
| Commercial assay or kit | Illumina MiSeq v2 Reagent Kit | Illumina | 15033625 | 2 × 250 cycles |
| Commercial assay or kit | Blue Pippin 1.5% agarose gel dye-free cassette | Sage Science | 250 bp - 1.5 kb DNA size range collections, Marker R2 | Target of 900 bp |
| Commercial assay or kit | Qubit dsDNA HS kit | Life Technologies | Q32854 | |
| Commercial assay or kit | PCR DIG Probe Synthesis Kit | Roche | 11636090910 | |
| Commercial assay or kit | Agilent 2100 Bioanalyzer High Sensitivity DNA Reagents | Agilent Technologies | 5067–4626 | |
| Chemical compound, drug | Fluconazole (FLC) | Alfa Aesar | J62015 | |
| Software, algorithm | MUMmer Sutie | *Kurtz et al., 2004* (DOI: 10.1186/gb-2004-5-2-r12) | v3.0 RRID:SCR_001200 | |
| Software, algorithm | Trimmomatic | *Bolger et al., 2014* (DOI: 10.1093/bioinformatics/btu170) | v0.33 RRID:SCR_011848 | |
| Software, algorithm | BWA | *Li, 2013* (DOI: 10.1093/bioinformatics/btp324) | v0.7.12 RRID:SCR_010910 | |
| Software, algorithm | Samtools | *Li et al., 2009* (DOI: 10.1093/bioinformatics/btp352) | v0.1.19 RRID:SCR_002105 | |
| Software, algorithm | Genome Analysis Toolkit | *McKenna et al., 2010* (DOI: 10.1101/gr.107524.110) | v3.4–46 RRID:SCR_001876 | |
| Software, algorithm | REPuter | *Kurtz et al., 2001* (DOI: 10.1093/nar/29.22.4633) | V1.0 https://bibiserv.cebitec.uni-bielefeld.de/reputer | |
| Software, algorithm | Yeast Analysis Mapping Pipeline | *Abbey et al., 2014* (DOI: 10.1186/s13073-014-0100-8) | v1.0 | |
| Software, algorithm | Graphpad Prism | https://www.graphpad.com | v6.0 RRID:SCR_002798 | |
| Software, algorithm | ImageJ | https://imagej.nih.gov/ij/? | v2.0.0-rc-30/1.49 s RRID:SCR_003070 | |
| Software, algorithm | Integrative Genomics Viewer | *Thorvaldsdóttir et al., 2013* (DOI: 10.1093/bib/bbs017) | v2.3.92 RRID:SCR_011793 | |
| Software, algorithm | R | https://www.r-project.org | v3.5.2 RRID:SCR_001905 | |
| Software, algorithm | Candida Genome Database | http://Candidagenome.org | RRID:SCR_002036 | |
| Other | Propidium Iodide | Invitrogen | P3566 | 25 ug/ml final concentration |
| Other | Ribonuclease A | MP Biomedicals | 101076 | 0.5 mg/ml final concentration |

### Yeast isolates and culture conditions

All isolates used in this study are shown in *Supplementary file 1*. Isolates were stored at −80°C in 20% glycerol. Isolates were grown at 30°C in YPAD (yeast peptone dextrose medium; *Rose W, 1990*) supplemented with 40 µg ml$^{-1}$ adenine and 80 µg ml$^{-1}$ uridine).

### In vivo evolution experiments

OPC isolates were obtained as previously described (*Forche et al., 2018*; *Solis and Filler, 2012*). Briefly, mice were orally infected with strain YJB9318 and single colony isolates were obtained from tongue tissue of mice on days 1, 2, 3, and 5 post infection and stored in 50% glycerol at −80°C for further use.

### In vitro evolution experiments

Six isolates were obtained from in vitro evolution experiments in the presence of antifungal drug (*Supplementary file 1*). Isolate AMS3053 was obtained on 20 µg/ml Miconazole agar plates as previously described (*Mount et al., 2018*). Isolates AMS3742, AMS3743, AMS3747, AMS3748, and AMS3744 were obtained from liquid batch culture evolution experiments conducted in 96-well format. Progenitor isolates were plated for single colonies on YPAD and incubated for 48 hr at 30°C. Single colonies were grown to saturation in liquid YPAD at 30°C. A 1:1000 dilution was made in YPAD medium containing either 1 µg/ml or 128 µg/ml of FLC. Plates were covered with BreathEAS-IER tape (Electron Microscope Science) and cultured in a humidified chamber for 72 hr at 30°C. At each 72 hr time point, cells were resuspended by pipetting and transferred into fresh media via a 1:1000 dilution and cultured for another 72 hr at 30°C, for 10 consecutive passages. After the final transfer, cells were immediately collected for genomic DNA isolation and ploidy analysis by flow cytometry.

To obtain AMS3743 isolates that had lost the i(4R) (*Figure 1—figure supplement 1*), 12 single colonies of AMS3743 were selected on YPAD plates at 30°C after 48 hr. All 12 single colonies had i(4R) (by PCR) and were used to initiate 12 YPAD-evolved lineages, each cultured for 24 hr in 4 ml liquid YPAD at 30°C with shaking. Every 24 hr, a 1:1000 dilution was inoculated into fresh YPAD medium. Cultures were passaged for 30 days. Cells from all 12 YPAD-evolved lineages were divided into tubes for −80°C storage, genomic DNA isolation, and CHEF analysis. All 12 YPAD-evolved lineages maintained i(4R) by CHEF analysis. CHEF gel densitometry analysis (see below) identified one lineage (AMS3743_10) that had a lighter i(4R) band density relative to the rest of the genome. AMS3743_10 was plated for single colonies on a YPAD plate and incubated at 30°C for 48 hr. Six single colonies were cultured for 24 hr in 4 ml liquid YPAD at 30°C with shaking, and cells were divided into tubes for −80°C storage, genomic DNA isolation, and CHEF analysis. One of the six single colonies lost the i(4R) (AMS3743_10_S6, *Figure 1—figure supplement 1*).

### Contour-clamped homogenous electric field (CHEF) electrophoresis

Samples were prepared as previously described (*Selmecki et al., 2005*). Cells were suspended in 300 µL 1.5% low-melt agarose (Bio-Rad) and digested with 1.2 mg Zymolyase (US Biological). Chromosomes were separated on a 1% Megabase agarose gel (Bio-Rad) in 0.5X TBE using a CHEF DRIII apparatus. Run conditions as follows: 60 s to 120 s switch, 6 V/cm, 120° angle for 36 hrs followed by 120 s to 300 s switch, 4.5 V/cm, 120° angle for 12 hrs.

### CHEF gel densitometry

Ethidium bromide stained CHEF gels were imaged using the GelDock XR imaging system (BioRad). Images were exported as .PNG files, converted to 32-bit, and analyzed using ImageJ (v2.0.0-rc-30/1.49 s). The total lane density (gray value, area under the curve) was collected for each sample. The density associated with i(4R) was determined by drawing a box around the i(4R) density peak (box distance was from each adjacent minimums). The fraction of i(4R) relative to the entire genome was determined by normalizing the i(4R) density relative to the total lane density. The population with lowest ratio of i(4R) relative to total genome (AMS3743_10) was used for single colony analysis.

### Southern hybridization

DNA from CHEF gels was transferred to BrightStar Plus nylon membrane (Invitrogen). Probing and detection of the DNA was conducted as previously described (*Selmecki et al., 2005*; *Selmecki et al., 2008*; *Selmecki et al., 2009*). Probes were generated by PCR incorporation of DIG-11-dUTP into target sequences following manufacturer's instructions (Roche). Primer pairs used in probe design are listed in *Supplementary file 7*.

### PCR

All primer sequences were designed to avoid heterozygous or SNP loci in the reference genome SC5314 and clinical isolates. Primers and primer sequences are found in *Supplementary file 7*. PCR conditions for i(4R) were as follows: 95°C for 3 min, followed by 32 cycles of 95°C for 30 s, 55°C for 30 s, 72°C for 5.5 min, and a final extension at 72°C for 10 min. The PCR conditions for the Chr4 inversion (*Figure 5*) were the same as above, except the annealing temperature was 53°C and the extension time was 3.25 min.

### Flow cytometry

Cells were prepared as previously described (*Todd et al., 2018*). Briefly, cells were grown to a density of $1 \times 10^7$ in liquid medium and gently spun down (500 x g) for 3 min. The supernatant was removed and cells were fixed with 70% (v/v) ethanol for at least 1 hr at room temperature. Cells were then washed twice with 50 mM sodium citrate and sonicated (Biorupter Fisher Science) for 10–15 s at 30% power to separate the cells. Following sonication, cells were centrifuged and resuspended with 50 mM sodium citrate and incubated for at least 3 hr at 37°C in 0.5 mg ml$^{-1}$ RNase A (MP Biomedicals) in 50 mM sodium citrate (Fisher Scientific). Cells were stained with 25 µg ml$^{-1}$ propidium iodide (Invitrogen) overnight in the dark at 37°C. Cells were sonicated for 5–10 s at 15% power, and 30,000 cells were analyzed on a ZE5 cell analyzer (BioRad). Data were analyzed in FlowJo (https://www.flowjo.com/solutions/flowjo/downloads) (v10.4.1).

### Growth curve analysis

Growth curves were determined using a BioTek Epoch plate reader. Culture medium included YPAD or YPAD +32 µg/ml FLC (Alfa Aesar) Approximately $5 \times 10^3$ cells were inoculated into 200 µl culture medium in a clear, flat bottomed 96-well plate (Thermo Scientific). The plate was incubated at 30°C with double orbital shaking at 256 rpm, and the OD$_{600}$ was measured every 15 min. Data were collected with Gen5 Software (BioTek) and exported to Microsoft Excel for downstream analysis. All growth curves were conducted in individual biological triplicate on separate days.

### Illumina whole genome sequencing

Genomic DNA was isolated with phenol chloroform as described previously (*Selmecki et al., 2006*). Libraries were prepared using the NexteraXT DNA Sample Preparation Kit following the manufacturer's instructions (Illumina). DNA fragments between 600 and 1200 bp were selected for sequencing using a Blue Pippin 1.5% agarose gel dye-free cassette (Sage Science). Library fragments were analyzed with a Bioanalyzer High Sensitivity DNA Chip (Agilent Technologies) and Qubit High Sensitivity dsDNA (Life Technologies). Libraries were sequenced using paired-end, $2 \times 250$ reads on an Illumina MiSeq (Creighton University). Adaptor sequences and low-quality reads were trimmed using Trimmomatic (v0.33 LEADING:3 TRAILING:3 SLIDINGWINDOW:4:15 MINLEN:36 TOPHRED33) (*Bolger et al., 2014*). Reads were mapped to the *Candida albicans* reference genome (A21-s02-m09-r08) obtained 7 of October 2015 from the *Candida* Genome Database website: http://www.candidagenome.org/download/sequence/C_albicans_SC5314/Assembly21/archive/ C_albicans_SC5314_version_A21-s02-m09-r08_chromosomes.fasta.gz). The reads were mapped using the Burrows-Wheeler Aligner MEM algorithm using default parameters (BWA v0.7.12) (*Li, 2013*). Duplicate PCR amplicons were removed using Samtools (v0.1.19) (*Li et al., 2009*), and reads were realigned around possible indels using Genome Analysis Toolkit's RealignerTargetCreator and IndelRealigner (-model USE_READS -targetIntervals) (v3.4–46) (*McKenna et al., 2010*). All WGS data have been deposited in the National Center for Biotechnology Information Sequence Read Archive database as PRJNA510147. Sequence data obtained from published datasets are noted in *Supplementary file 1*.

## Identification of aneuploidy and copy number breakpoints

Preliminary identification of chromosomes containing CNVs was conducted using Illumina whole genome sequence data and the Yeast Mapping Analysis Pipeline (YMAP v1.0). Fastq files were uploaded to YMAP and read depth was plotted as a function of chromosome location using the reference genome *Candida albicans* (A21-s02-mo8-r09), with correction for chromosome end bias and GC content (*Abbey et al., 2014*). The average normalized genome coverage was determined for 45.5 kb non-overlapping windows across each chromosome using the YMAP GBrowse CNV track. The largest absolute difference between the average normalized genome coverage of two consecutive 45.5 kb windows was identified. To further refine CNV breakpoints, fastq files were aligned to the reference genome as above (Illumina Whole Genome Sequencing), read depth was calculated for every base pair in the nuclear genome using Samtools (samtools depth -aa) (v0.1.19), and normalized by read depth of the total nuclear genome using R (v3.5.2). The two consecutive 45.5 kb windows were further sub-divided into 5 kb windows. The average normalized read depth was determined for these 5 kb windows and a rolling mean of every two consecutive 5 kb windows was determined. CNV breakpoint boundaries were identified when 75% of four consecutive means had an average normalized read depth that deviated from the average normalized nuclear genome read depth by more than 25% in tetraploids or 50% in diploids (*Ford et al., 2015*). Boundaries were confirmed by visual inspection in Integrative Genomics Viewer (IGV v2.3.92) (*Thorvaldsdóttir et al., 2013*). CNV breakpoints were then determined using visual inspection of total read depth and allele ratio analysis (when the breakpoint was surrounded by heterozygous sequence) within unique, non-repeat sequences. CNV breakpoint positions were compared to *Supplementary file 2* and breakpoints were assigned a repeat name if they fell within 2 kb of a long repeat sequence.

## Enrichment of CNV breakpoints at long repeat sequences

Enrichment analysis of CNV breakpoints was conducted using a two-tailed Fisher's Exact Test in Bedtools (Bedtools v2.28.0) with default parameters (*Quinlan and Hall, 2010*). Briefly, two bed files were generated with 1) the start and stop positions of all long repeat sequences and 2) the start and stop positions of all long repeat sequences located within 2 kb of a CNV breakpoint (*Supplementary file 2*, excluding the complex tandem repeat genes). The overlap of observed breakpoints and long repeat sequences was compared to the expected overlap between CNV breakpoints and long repeat sequences, given the total genome coverage of long repeat sequences. The minimum overlap required was a single base pair between a CNV breakpoint and repeat sequence.

## Identification of long-range homozygosity breakpoints

Illumina whole genome sequence data were analyzed using YMAP (v1.0) and IGV (v2.3.92). First, fastq files were uploaded to YMAP and the density of heterozygous SNPs was determined for non-overlapping 5 kb windows and plotted by chromosomal position in standard SNP/LOH view (default parameters, baseline ploidy was 2N for all isolates except AMS3420, which was 4N). Approximate positions of all long-range homozygous and heterozygous transitions were determined within 20–25 kb. To further refine LOH breakpoints, fastq files were aligned to the reference genome as above (Illumina Whole Genome Sequencing) and visualized in IGV. All heterozygous to homozygous (and vice versa) transitions were recorded when four or more consecutive loci were heterozygous and transitioned to four or more homozygous loci (and vice versa). The minimum distance covered by the four or more consecutive loci was greater than 300 bp and all four of the loci were located within unique, non-repeat sequences. Additionally, all heterozygous loci utilized for breakpoint analysis had an alternate allele frequency greater than or equal to 20%, read depth greater than 10 reads, and both forward and reverse strands that supported the alternate allele (*Selmecki et al., 2015*). The breakpoints of these long-range homozygous tracks ('LOH breakpoints') were recorded as the last heterozygous locus and the first homozygous locus of the heterozygous > homozygous transition, and vice versa for the homozygous > heterozygous transition. Long-range LOH breakpoints were then compared to *Supplementary file 2* and were assigned a repeat number if they fell within 2 kb of a long repeat sequence (*Supplementary file 4*).

## Identification of inversion breakpoints

Additional positions of predicted chromosomal inversions were obtained from *Hirakawa et al. (2015)*, Table S13. Coordinates corresponding to potential inversions were obtained using Break-Dancer or NUCmer (*Hirakawa et al., 2015*). The distance between the BreakDancer or NUCmer coordinates (start and stop) and the nearest long repeat sequence was determined. If a long repeat sequence occurred within 2 kb of either BreakDancer or NUCmer coordinates, the repeat number and family were recorded. Disagreement between BreakDancer and NUCmer coordinates that coincided with breakpoints in different repeat families (representing more complex chromosome rearrangements or inversions) were removed from the analysis. Additionally, all NUCmer or Breakdancer positions that occurred within *ALS* gene family repeats were removed from the analysis because the BreakDancer and NUCmer coordinates did not support a consistent length of sequence inversion (likely due to mapping errors within and between *ALS* repeats). The long repeat sequences identified at these potential inversion breakpoints, including those shared across different isolates, are summarized in *Supplementary file 6*.

## Microsatellite repeat identification

Short repetitive sequences found at either copy number breakpoints or allele ratio breakpoints were analyzed using REPuter (*Kurtz et al., 2001*) with a minimum repeat length of 8 bp. Analysis was conducted using the forward, reverse, complement, and palindromic match direction.

## Identification of long repeat sequences

Repeat sequences within the *C. albicans* genome were identified using the MUMmer suite (v3.0) (*Kurtz et al., 2004*). Whole genome sequence alignment with NUCmer (nucmer –maxmatch –nosimplify) identified all maximum-length matches with 100% sequence identity (minimum match length of 20 bp) within the *Candida albicans* SC5314 reference genome (A21-s02-m09-r08, obtained 7 of October 2015 from the *Candida* Genome Database (CGD): http://www.candidagenome.org/download/sequence/C_albicans_SC5314/Assembly21/archive/ C_albicans_SC5314_version_A21-s02-m09-r08_chromosomes.fasta.gz). All maximum length matches were identified, regardless of their uniqueness (meaning all matches in the genome were identified). Then, all sequence matches were clustered and extended to obtain a maximum-length colinear string of matches if they were separated by no more than 90 nucleotides (NUCmer default parameters). Three repeat matches shared less than 80% sequence identity, therefore an 80% cutoff was used for the final long repeat analysis (*Supplementary file 2*), similar to previous studies (*Achaz et al., 2000*; *Warren et al., 2014*). All sequences that self-aligned to the same genomic position were removed.

Repeat matches were annotated using the reference genome feature file (C_albicans_SC5314_version_A21-s02-m09-r08_Chromosomal_feature file) and repeat tracks obtained from CGD (*Skrzypek et al., 2017*). To highlight uncharacterized long repeat sequences, repeats associated with the three major classes of repetitive DNA in *C. albicans* were removed, including the rDNA locus, MRS sequences (*RPS*, *HOK*, and *RB2*), telomere-proximal regions, as well as ambiguous sequences (containing poly-N nucleotides). These regions are highly variable and difficult to analyze with short-read sequencing techniques (*Chibana et al., 2000*; *Chibana et al., 1994*; *Chindamporn et al., 1998*; *Goodwin and Poulter, 2000*; *Hoyer and Cota, 2016*; *Hoyer et al., 1995*; *Levdansky et al., 2008*). Telomere-proximal regions were determined as the region from each chromosome end to the first confirmed, non-repetitive-genome feature, similar to previous studies (*Ene et al., 2018*; *Hirakawa et al., 2015*): Chr1: 1–10000, Chr1:3181000–3188548, Chr2: 1–5000, Chr2: 2228650–2232035, Chr3: 1–15000, Chr3: 1787000–1799406, Chr4: 1–2700, Chr4: 1597200–1603443, Chr5: 1–3800, Chr5: 1183000–1190928, Chr6: 1–3000, Chr6: 1031500–1033530, Chr7: 1–75, Chr7: 942300–949616, ChrR: 1–4500, ChrR: 2286355–2286389. Telomere-associated genes, including *TLO* genes, that were not positioned in these telomere-proximal regions were maintained.

All long repeat sequences were verified using BLAST and IGV. Repeat copies that were on the same chromosome were defined as either tandem, mirrored, or inverted using the repeat start and end positions obtained from NUCmer and manually inspected in IGV. Tandem repeat sequences are in the same orientation on the same strand, mirrored repeat sequences are in opposite orientations

on the same strand, and inverted repeat sequences are in opposite orientations on the opposite strand. Spacer length was obtained by calculating the shortest distance between repeat matches.

After the post-alignment annotations and filtration, repeats were combined into repeat families if they shared an identical match. For example, if repetitive sequence A was matched with sequence B, sequences A and B were combined into one family. In some instances, a sequence matched with more than one sequence (e.g. A matched with B and C). In these cases, all matched sequences were combined into one family. In total, 230 repeat families were identified with sequence identities of $\geq$80% (median value of 92.9%) between all copies of the repeat within a family. Of these 230 families, 68 included more than two copies per genome (*Supplementary file 2*).

The fraction of the genome covered by long repeat sequences was determined by multiplying the average copy length of each repeat family by the number of copies of that repeat family found throughout the genome (excluding the complex tandem repeat genes). The sum of the average copy length of all repeat families (409129 bp) was then divided by the length of the haploid *Candida albicans* SC5314 reference genome (excluding the mt-DNA, 14280189 bp) to determine that 2.87% of the genome is covered by long repeat sequences (*Figure 2—source data 1*).

## Annotation of repeat sequences

The long repeat sequences were annotated according to the genomic features contained within each matched repeat sequence using the *C. albicans* genome feature file described above. The genomic features included were: lone long terminal repeats (LTRs) lacking ORFs, retrotransposons, tRNAs, ORFs, and intergenic sequences. Repeat matches containing ORFs included partial ORF sequences, single complete ORF sequences, and multiple ORFs and intergenic sequences. In cases where one repeat copy contained a genome feature, but the other repeat copy contained an intergenic sequence (no genome feature), this latter repeat was flagged as 'Unannotated Intergenic Sequence' and both repeat copies were assigned the feature found at the annotated repeat copy (*Supplementary file 2*). All unannotated sequences were verified in both V21 and V22 of the *C. albicans* reference genome (*Skrzypek et al., 2017*).

Of the known LTRs present within the *C. albicans* genome, only five were not detected in the MUMmer analysis. Analysis of the five undetected LTRs using BLASTN revealed that they lacked an exact match of 20 nucleotides required to establish a matched repeat pair.

All full-length ORF coding sequences within the *C. albicans* reference genome (C_albicans_SC5314_version_A21-s02-m09-r08_chromosomes.fasta.gz) were analyzed for length and GC content using EMBOSS Infoseq (http://imed.med.ucm.es/cgi-bin/emboss.pl?_action=input&_app=infoseq). All full-length ORF coding sequences were divided into coding sequences that were contained within long repeat sequences or coding sequences that were not contained within long repeat sequences (excluding the complex tandem repeat genes, *Supplementary file 2*, *Figure 2—figure supplement 3D and E*). If a long repeat sequence contained a partial ORF sequence, the full-length coding sequence was used in the analysis. Similarly, if a long repeat sequence contained multiple ORF sequences, the full-length coding sequence of each ORF were included in the analysis.

## Exclusion of complex tandem repeat genes

Five ORFs and one gene family with known, complex embedded tandem repeats were confirmed by NUCmer (*PGA18*, *PGA55*, *EAP1*, Adhesin-like *orf19.1725*, *CSA1*, and the *ALS* gene family comprised of eight ORFs, *Supplementary file 2*) (*Levdansky et al., 2008*; *Wilkins et al., 2018*). Assignment of a genome copy count was not possible for these tandem repeat genes due to the extreme complexity of matched repeat sequences. For this reason, all repeat copy counts and analysis using copy counts exclude the complex tandem repeat genes listed above and are indicated throughout the text (*Supplementary file 2*).

## Statistical analyses

For this study, biological replicates are defined as a single, independent culture derived from a frozen −80°C glycerol stock. Data were analyzed using GraphPad Prism v6 and made into graphical representations using RSudio v1.1.463. All p-values below 0.05 were considered significant.

## Acknowledgements

We thank all members of the Selmecki laboratory, especially Curtis Focht, Alison Guyer, Robert Thomas, and Annette Beach for technical assistance. We thank Dr. Robin Dowell, Dr. Mary Ann Allen, and Dr. Hung-Ji Tsai for feedback on the manuscript and helpful discussions. Support for this research was provided by LB692 NE Tobacco Settlement Biomedical Research Development New Initiative Grant (to AS), NE Established Program to Stimulate Competitive Research (EPSCoR) First Award (to AS), NE Department of Health and Human Services (LB506-2017-55) award (to AS), CURAS Faculty Research Fund Award (to AS), and NIH-NCRR COBRE grant P20RR018788 sub-award (to AS). AF was supported by NIH grant R15 AI090633. The sequencing datasets generated during this study are available in the Sequence Read Archive repository under project PRJNA510147.

## Additional information

### Funding

| Funder | Grant reference number | Author |
|---|---|---|
| Nebraska LB692 New Initiatives Grants | LB692 NE Tobacco Settlement Biomedical Research Development New Initiative Grant | Anna Selmecki |
| Nebraska's Established Program to Stimulate Competitive Research | EPSCoR First Award | Anna Selmecki |
| Nebraska Department of Health and Human Services | LB506-2017-55 | Anna Selmecki |
| Creighton University | CURAS Faculty Faculty Research Fund | Anna Selmecki |
| National Center for Research Resources | P20RR018788 sub award | Anna Selmecki |
| National Institutes of Health | R15 AI090633 | Anja Forche |

The funders had no role in study design, data collection and interpretation, or the decision to submit the work for publication.

### Author contributions

Robert T Todd, Conceptualization, Data curation, Formal analysis, Supervision, Funding acquisition, Validation, Investigation, Visualization, Methodology, Writing—original draft, Writing—review and editing; Tyler D Wikoff, Data curation, Formal analysis, Validation, Investigation, Visualization, Methodology; Anja Forche, Resources, Data curation, Funding acquisition, Validation, Writing—original draft, Writing—review and editing; Anna Selmecki, Conceptualization, Resources, Data curation, Formal analysis, Supervision, Funding acquisition, Validation, Investigation, Methodology, Writing—original draft, Writing—review and editing

### Author ORCIDs

Robert T Todd [iD] https://orcid.org/0000-0002-4522-7124
Anja Forche [iD] https://orcid.org/0000-0002-3004-5176
Anna Selmecki [iD] https://orcid.org/0000-0003-3298-2400

### Decision letter and Author response

Decision letter https://doi.org/10.7554/eLife.45954.039
Author response https://doi.org/10.7554/eLife.45954.040

## Additional files

**Supplementary files**

• Supplementary file 1. Strains used in this study.
DOI: https://doi.org/10.7554/eLife.45954.022

• Supplementary file 2. Long repeat sequences in the *Candida albicans* genome.
DOI: https://doi.org/10.7554/eLife.45954.023

• Supplementary file 3. Copy number variation breakpoints in diverse *C. albicans* isolates.
DOI: https://doi.org/10.7554/eLife.45954.024

• Supplementary file 4. Loss of heterozygosity breakpoints in diverse *C. albicans* isolates.
DOI: https://doi.org/10.7554/eLife.45954.025

• Supplementary file 5. Location of telomere-seed sequences throughout the *C. albicans* genome.
DOI: https://doi.org/10.7554/eLife.45954.026

• Supplementary file 6. Predicted inversion breakpoints in diverse *C. albicans* isolates.
DOI: https://doi.org/10.7554/eLife.45954.027

• Supplementary file 7. Primers used in this study.
DOI: https://doi.org/10.7554/eLife.45954.028

• Transparent reporting form
DOI: https://doi.org/10.7554/eLife.45954.029

**Data availability**

All data generated and analyzed during this study are included in the manuscript and supporting files. Source data files have a been provided for Figure 1, Figure 1—figure supplement 1, Figure 2, Figure 2—figure supplement 2, Figure 2—figure supplement 3, Figure 6, and Figure 6—figure supplement 1. All genomic data are deposited in SRA under accession PRJNA510147.

The following dataset was generated:

| Author(s) | Year | Dataset title | Dataset URL | Database and Identifier |
|---|---|---|---|---|
| Todd RT, Forche A, Selmecki A | 2019 | Segmental Aneuploidies in Candida albicans | https://www.ncbi.nlm.nih.gov/bioproject/PRJNA510147 | NCBI SRA, PRJNA510147 |

The following previously published datasets were used:

| Author(s) | Year | Dataset title | Dataset URL | Database and Identifier |
|---|---|---|---|---|
| Hirakawa MP, Martinez DA, Sakthikumar S, Anderson MZ, Berlin A, Gujja S, Zeng Q, Zisson E, Wang JM, Greenberg JM, Berman J, Bennett RJ, Cuomo CA | 2013 | Candida albians Umbrella Comparative genomics project | https://www.ncbi.nlm.nih.gov/bioproject/?term=PRJNA193498 | NCBI SRA, PRJNA193498 |
| Ford CB, Funt JM, Abbey D, Issi L, Guiducci C, Martinez DA, Delorey T, Li BY, White TC, Cuomo C, Rao R, Berman J, Thompson D, Regev A | 2014 | The evolution of gradual acquisition of drug reisstance in clinical isolates of Candida albicans | https://www.ncbi.nlm.nih.gov/bioproject/?term=PRJNA257929 | NCBI SRA, PRJNA257929 |
| Ropars J, Maufrais C, Diogo D, Marcet-Houben M, Perin A, Sertour N, Mosca K, Permal E, Laval G, Bouchier C, Ma L, Schwarts | 2018 | Whole Genome sequencing of 182 isolates of the fungal pathogen of humans Candida albicans | https://www.ncbi.nlm.nih.gov/bioproject/432884 | NCBI SRA, PRJNA432884 |

K, Voelz K, May RC,
Poulain J, Battail C,
Wincker P, Borman
AM, Chowdhary A,
Fan S, Kim SH, Le
Pape P, Romeo O,
Shin JH, Gabaldon
T, Sherlock G,
Bougnoux M,
d'Enfert C

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
