## [Decision Letter]

Thank you for submitting your article "Genome plasticity in *Candida albicans* is driven by long repeat sequences" for consideration by *eLife*. Your article has been reviewed by three peer reviewers, one of whom is a member of our Board of Reviewing Editors, and the evaluation has been overseen by Detlef Weigel as the Senior Editor. The following individual involved in review of your submission has agreed to reveal their identity: Matthew Anderson (Reviewer #2).

The reviewers have discussed the reviews with one another and the Reviewing Editor has drafted this decision to help you prepare a revised submission.

We appreciate how your study details the role of repetitive sequences in the *C. albicans* genome on production of genetic variation, which is a significant step towards understanding how genetic variation is produced in the plastic *C. albicans* genome. More specifically, they show that changes in copy number and loss of heterozygosity associate with the repeats, which often exist as multi-copy sequences found at a range of distances across the genome. Repeats covering ORFs are enriched for LOH, CNV breakpoints, and inversions.

As you can see in the individual reviewer's reports (below), all reviewers agree that your study is solid and interesting. After discussing the reviews among the reviewers and editors, we suggest the following essential changes to the manuscript.

1) We think it is important to add a more detailed description (in the Materials and methods section) and critical discussion (in the Results or Discussion section) on how the repeats were mapped starting from short-read sequences.

2) We suggest more elaborate statistical testing and/or a clearer description of what exactly is tested, and how, to assess the significance of repeat enrichment.

While the other issues and suggestions raised by the individual reviewers were not deemed crucial, we suggest you still consider adapting the manuscript accordingly.

*Reviewer #1:*

This study investigates the occurrence and sequence context of structural variation in the *Candida albicans* genome. The authors find several instances of inversions, deletions, LOH and translocation events that are associated with repeat sequences spread across the *C. albicans* genome, including centromeric and telomeric repeats, as well as repeats in ORFs. The breakpoints of these events were often at regions of higher and longer sequence identity.

Overall, this is a nice comprehensive study describing the importance of repeated sequences in genome plasticity.

My only concern is that the study does perhaps not offer much novel biological insight – repeats / regions with high sequence identity have already often been reported to form the breakpoints of structural variation in genomes. That said, the strength of this paper is that it gives a more comprehensive view on the phenomenon, which in itself has merit.

*Reviewer #2:*

The manuscript by Todd et al. details the role of repetitive sequences in the *C. albicans* genome on production of genetic variation, which is a significant step towards understanding how genetic variation is produced in the plastic *C. albicans* genome. More specifically, they show that changes in copy number and loss of heterozygosity associate with the repeats, which often exist as multi-copy sequences found at a range of distances across the genome. Repeats covering ORFs are enriched for LOH, CNV breakpoints, and inversions.

The inverted CEN4 breaks apart the CENP-A binding site. It would be interesting to know if CENP-A still binds the fragment of the binding region not disrupted during inversion in Chr4B and which homolog (A or B) built the i(4R) chromosome. This has implications in the potential for future recombination and accurate segregation of the i(4R), which, as the authors noted, is quite high compared to other trisomies. While it is stated in the Discussion that knowing this could be of interest in the future, it has implication on the current study.

A statistical test to show enrichment of CNBs within repeats would be helpful when introduced. While it is expected there to be enrichment, if repeat regions span a significant portion of the genome, 13 events may not be sufficient to see enrichment. A CNB between two repeats spaced by 70 kb is not particularly unexpected if when taking into consideration the distance between all repeats begins to approach the full genome size. What may help the reader see the association of the CNB better to the repeats themselves is to zoom in to ~nucleotide resolution and using a sliding window to show that the general copy number changes occur over the repeat as would be expected if they are involved in the recombination itself. Figure 3C does this well but Figure 3B does not. It is difficult to discern any of this from Figure 3—figure supplement 1. An amalgamated panel of all CNBs or LOH relative to their repeat may be best to summarize the findings concisely.

CNBs such as that displayed for AMS3053 on Chr3L that occur across long repeated sequenced with very high (99+%) identity would be hard to map by Illumina short-read sequencing. This is seen somewhat in the IGV snapshot where the repeat regions have an increase in read coverage compared to the internal unique sequence. It would worth including long-range sequencing (MinION or otherwise) for a few select events such as this to demonstrate that the proposed rearrangements are reflected in contiguous pieces of DNA that can span these repeats. Additionally, if these repeats contain genes and are 99+% identical, are the CDS within these regions similarly identical, indicating parologous gene duplications?

Segmental duplications including centromeres are unexpected as this may promote chromosome instability by including multiple kinetochore attachments on the same DNA molecule. Give that 2 strains contained these or their novelty; it would be worth testing if chromosome segregation is distorted in these strains as a result of centromere duplication. Alternatively, one may be activated, which could be tested by CENP-A ChIP-PCR. These events should be tied more closely to the SSA mechanism described in the Discussion.

A critical piece of information missing from the Materials and methods is how reads that could be mapped to multiple places were dealt with during alignment. As some repeats are 99+% identical, it would be hard to map those regions uniquely. The spike in heterozygosity at repeats could be due to random assignment of reads to one or the other repeat resulting in a het call at a homozygous position for each.

The selective pressures promoting retention of segmental deletions are interesting as these are often expected to have greater deleterious consequences than segmental amplifications. While not necessary, it would be helpful to know the fitness consequences of this deletion in the context of OPC in which AMS3420 or CEC2871 was obtained, a bloodstream model of infection, or a commensal colonization model. In short, why would loss of HGT1 and HGT2 benefit the cell during infection enough to be observed?

Breakpoints removed from *ALS* genes may be due less to poor mapping than rearrangements. How similar was the frequency of called breakpoints in comparable regions encoding tri- or di-nucleotide repeats as are found in the *ALS* sequences?

Are there features that distinguish between repeat-rich (Chr3R) and repeat-poor (Chr7L) chromosome arms? E.g., GC content, gene density, UTR length, etc., this will be begin to provide predictive correlates to repeats and recombination potential.

*Reviewer #3:*

In this paper Todd et al. analyse of the role that multi-copy genes play in generating structural variation in the *Candida albicans*genome. Through a comprehensive annotation of the reference genome and an analysis of structural variation in "evolved" strains the authors convincingly show that various repetitive elements have created genomic variation, some of which are associated with adaptive traits.

Overall, the authors have used appropriate methods, drawn reasonable conclusions and produced a well-written manuscript. I think the results will be an important contribution to the study of *C. albicans* in particular and genome evolution in general. I do have a number of small issues that I think could improve the paper, which I detail below.

Much is made in both the Results and Discussion section about thelarge 'spacer distance' between intra-chromosomal repeats (e.g. subsection “Identification of long repeat sequences throughout the *C. albicans* genome”, second paragraph). This is certainly an interesting result, and the raw data makes it clear this is a real phenomenon. However, I think the manuscript could do with some more clarity about:

a) Why this statistic is of interest;b) Precisely what hypothesis is being tested in this "1-way ANOVA withposttest…"

I suggest a sentence in the Results section describing the motivation forcalculating this distance. I am not sure what we are mean to glean from the factspacer-distance is not (significantly) correlated with chromosome size (given the small number of chromosomes). It would be good to make the biological motivation for this test explicit or reconsider the test. If the implication is that the repeat-copies are approximately uniformly distributed across chromosomes then a statistical test for this (rather than trend with chromosome size) may be a better test? Alternatively, it may be helpful to simply visualize the distribution of spacer-sizes in each chromosome via a histogram or 1-D kernel estimate).

I found myself being slightly tripped up by terminology re LTRs andretrotransposons. Presumably, the large number of repeats identified as "LTRS"

(e.g. subsection “Identification of long repeat sequences throughout the *C. albicans* genome”, last paragraph) are long terminal repeats that lack ORFs (non-autonomous or "lone" LTRs) while the small number of retrotransposons will include complete LTR retrotransposons. Perhaps a statement or edit to the Results section making this clear will help readers.

---

## [Author Response]

As you can see in the individual reviewer's reports (below), all reviewers agree that your study is solid and interesting. After discussing the reviews among the reviewers and editors, we suggest the following essential changes to the manuscript.1) We think it is important to add a more detailed description (in the Materials and methods section) and critical discussion (in the Results or Discussion section) on how the repeats were mapped starting from short-read sequences.

Thank you for this suggestion. Substantial details were added to the Materials and methods section (subsections “Identification of Aneuploidy and Copy Number Breakpoints”, “Identification of Long-Range Homozygosity Breakpoints”, “Identification of Inversion Breakpoints” and “Identification of Long Repeat Sequences”) regarding how we mapped the initial repeat positions from the reference genome assembly (Fasta file), and how we mapped all breakpoints (CNV, LOH, and inversions) using Illumina short-read sequence data. As we discuss below for reviewer 2, these breakpoints were defined within 2 kb of repeat sequences and were supported by unique, non-repeat sequences, since short-reads within the repeat sequences can map with high quality to multiple places in the genome. Analysis of short-read datasets (including previously published datasets) highlights the importance of this approach in identifying genomic features that are driving structural and allelic variation across *C. albicans* and future genome datasets. A critical discussion of short-read sequencing was added to the Discussion (subsection “Inverted repeat sequences directly associated with the CENP-A-binding centromere core sequences facilitate isochromosome formation”, last paragraph).

2) We suggest more elaborate statistical testing and/or a clearer description of what exactly is tested, and how, to assess the significance of repeat enrichment.

We re-evaluated all of the statistical methods and added new statistical analyses of both existing data and newly generated figures. All raw data and statistical analyses were updated in the source data files (Figure 2—source data 1-4 and Figure 6—source data 1). We included the null hypothesis for all statistical tests within the source data files and provided a clearer description of each test within the main text. We included tests of distributions and post hoc analyses, including Kolmogorov-Smirnov test and Kruskal-Wallis test with Dunn’s multiple comparison (Figure 2—figure supplement 2B, Figure 2—figure supplement 3A-E, Figure 6A and B, and Figure 6—figure supplement 1A and B). For example, to visualize the distribution of spacer lengths between repeat matches on each chromosome we generated new Figure 2—figure supplement 2B and Figure 2—source data 2, and found there was a significant difference in the distribution of spacer lengths across all chromosomes (p < 0.035, Kruskal-Wallis test with Dunn’s multiple comparison). Finally, we determined the fraction of the genome covered by long repeat sequences and assessed the significance of repeat enrichment using Bedtools (new Materials and methods subsection “Enrichment of CNV Breakpoints at Long Repeat Sequences”). There was significant enrichment for CNV breakpoints within long repeat sequences (p < 0.0001, two-tailed Fishers Exact Test).

While the other issues and suggestions raised by the individual reviewers were not deemed crucial, we suggest you still consider adapting the manuscript accordingly.

Below we include a point-by-point response to the individual reviewer’s comments. We thank the Editors and reviewers for helping us improve the final manuscript.

Reviewer #2:[…] The inverted CEN4 breaks apart the CENP-A binding site. It would be interesting to know if CENP-A still binds the fragment of the binding region not disrupted during inversion in Chr4B and which homolog (A or B) built the i(4R) chromosome. This has implications in the potential for future recombination and accurate segregation of the i(4R), which, as the authors noted, is quite high compared to other trisomies. While it is stated in the Discussion that knowing this could be of interest in the future, it has implication on the current study.

We agree that the impact of the *CEN4* inversion on CENP-A binding and chromosome stability is an exciting future direction. To clarify, we do not know which SC5314 *C. albicans* homolog (A or B) is inverted. Previously, we used Chr4A and Chr4B in Figure 1S to distinguish between alleles, but these labels have been replaced with Homolog 1 and 2.

Unfortunately, we cannot distinguish between the two different centromere alleles on the i(4R) because the recombination event between the *CEN4* repeats (forming i(4R)) results in a hairpin structure. A PCR primer in unique Chr4R sequence will amplify both orientations of the *CEN4* inter-repeat sequence from an i(4R). The two options are drawn below, where Chr4R sequence is distal to the red arrows in both.

A statistical test to show enrichment of CNBs within repeats would be helpful when introduced. While it is expected there to be enrichment, if repeat regions span a significant portion of the genome, 13 events may not be sufficient to see enrichment. A CNB between two repeats spaced by 70 kb is not particularly unexpected if when taking into consideration the distance between all repeats begins to approach the full genome size.

We thank the reviewer for this suggestion. First, one point of clarification: there are 26 copy number variation breakpoints in 13 isolates. The fraction of the genome covered by long repeat sequences was 2.87% (409129 bp/14280189 bp), excluding the complex tandem repeat genes. This calculation was added to the Results (subsection “Identification of long repeat sequences throughout the *C. albicans* genome”, first paragraph) and Materials and methods sections (subsection “Identification of Long Repeat Sequences”, last paragraph). Again, all CNV breakpoints occurred within 2 kb of these long repeat sequences. Therefore, we performed a Fisher’s Exact Test to determine the probability that a breakpoint in the genome would overlap with a long repeat sequence (p < 0.0001, Fisher’s Exact Test). Given the small p-value, we can reject the null hypothesis that long repeat sequences and observed breakpoints are independent of one another. The manuscript has been updated as follows: “Strikingly, every CNV breakpoint occurred within 2 kb of a long repeat sequence, ranging from 248 bp to ~4.76 kb in length. Observed breakpoints had significantly more overlap with long repeat sequences than expected given the total genome coverage of long repeat sequences (p < 0.0001, two-tailed Fishers Exact Test, See Materials and methods).” Additionally, a new Materials and methods subsection, “Enrichment of CNV Breakpoints at Long Repeat Sequences” was added for this analysis.

What may help the reader see the association of the CNB better to the repeats themselves is to zoom in to ~nucleotide resolution and using a sliding window to show that the general copy number changes occur over the repeat as would be expected if they are involved in the recombination itself. Figure 3C does this well but Figure 3B does not. It is difficult to discern any of this from Figure 3—figure supplement 1. An amalgamated panel of all CNBs or LOH relative to their repeat may be best to summarize the findings concisely.

We agree these data are complex, however nucleotide resolution in a single figure is challenging given the repeat length and spacer lengths of many of the repeats. We intended Figure 3—figure supplement 1 to highlight the size and structure of diverse repeats found on each chromosome (1-R), highlighting repeat sequences that contain multiple ORFs and intergenic sequences. Nucleotide resolution is provided in Figure 3B and C and Figure 4B and C to support that all breakpoints are occurring within 2 kb of a repeat sequence. Additionally, we updated Figure 3B to better show copy number changes across this region. In doing so, the allele ratios are now less obvious, but the conclusion remains the same.

CNBs such as that displayed for AMS3053 on Chr3L that occur across long repeated sequenced with very high (99+%) identity would be hard to map by Illumina short-read sequencing. This is seen somewhat in the IGV snapshot where the repeat regions have an increase in read coverage compared to the internal unique sequence. It would worth including long-range sequencing (MinION or otherwise) for a few select events such as this to demonstrate that the proposed rearrangements are reflected in contiguous pieces of DNA that can span these repeats.

We agree that future experiments using long-range sequencing will be helpful to better assemble the reference genome and to assemble the molecules de novo that resulted from recombination in these clinical isolates. A new paragraph was added to the Discussion section to address this point (subsection “Inverted repeat sequences directly associated with the CENP-A-binding centromere core sequences facilitate isochromosome formation”, last paragraph). However, we do not think that this is necessary to support the current findings – the observations and conclusions are supported by short read sequence data and the reference genome sequence. Indeed, part of the novelty of the current study is that these repeat sequences and breakpoints were identified using many published Illumina datasets.

Additionally, if these repeats contain genes and are 99+% identical, are the CDS within these regions similarly identical, indicating parologous gene duplications?

Yes, ORF CDSs contained within a long repeat sequence shared a similarly high sequence identity with the entire repeat (subsection “Identification of long repeat sequences throughout the *C. albicans* genome”, fifth paragraph). Pairwise sequence analyses (EMBOSS WATER) for repeats containing multiple ORFs and single complete ORFs supported that these CDSs are paralogs resulting from duplication events. For example, repeat family 124 (99.53% sequence identity), contains four ORFs and intergenic sequences. The paired ORF CDSs shared similarly high sequence identity (99.3%, 99.8%, 99.5%, 99.7%, respectively) supporting paralogous duplication of multiple contiguous ORFs. Repeats containing single complete ORFs also had similarly high sequence identity between the entire repeat and the CDS contained within the repeat. For example, repeat family 52 had a sequence identity of 99.57%, while the CDSs had sequence identity (99.8%) that was even higher than the entire repeat.

Additionally, we now clearly state that repeat sequences that contain single complete ORF sequences or multiple ORFs and intergenic sequences are paralogs resulting from duplication events. These statements have been added to the Introduction, Results and figure legends (Introduction, last paragraph, subsection “Identification of long repeat sequences throughout the *C. albicans* genome”, fourth paragraph and Figure 2—figure supplement 1 legend).

Segmental duplications including centromeres are unexpected as this may promote chromosome instability by including multiple kinetochore attachments on the same DNA molecule. Give that 2 strains contained these or their novelty, it would be worth testing if chromosome segregation is distorted in these strains as a result of centromere duplication. Alternatively, one may be activated, which could be tested by CENP-A ChIP-PCR. These events should be tied more closely to the SSA mechanism described in the Discussion.

Thank you for this suggestion. We have tied these examples more clearly to the proposed SSA mechanisms described in the Discussion (subsection “DNA double-strand breaks are repaired using long repeat sequences found across the *C. albicans* genome”, third paragraph). However, we think that a detailed characterization of centromere silencing and dynamics of chromosome stability in specific isolates is beyond the scope of this current manuscript.

A critical piece of information missing from the Materials and methods is how reads that could be mapped to multiple places were dealt with during alignment. As some repeats are 99+% identical, it would be hard to map those regions uniquely. The spike in heterozygosity at repeats could be due to random assignment of reads to one or the other repeat resulting in a het call at a homozygous position for each.

Thank you for suggesting clarification of these details. All reads were mapped to the reference genome with the same parameters (Materials and methods subsection “Illumina Whole Genome Sequencing”). We agree that because Illumina short-reads within the repeat sequence can map, with high quality, to multiple positions in the genome, they cannot be used for allele ratio analyses. For this reason, unique, non-repeat sequences were used to determine breakpoint positions and all breakpoints were thus defined as occurring within 2 kb of a long repeat sequence. We have updated the Materials and methods section with a more detailed explanation of all breakpoint analyses (subsections “Identification of Aneuploidy and Copy Number Breakpoints”, “Identification of Long-Range Homozygosity Breakpoints” and “Identification of Inversion Breakpoints”). Additionally, we revised the Results section on heterozygosity islands caused by repeat sequences to better highlight our findings. This section now includes: “As expected, levels of heterozygosity were high within long repeat sequences due to the ability of short-read (Illumina) sequences to map to multiple positions in the genome (e.g. the heterozygous bases within repeat sequences in Figure 4B and C)”.

The selective pressures promoting retention of segmental deletions are interesting as these are often expected to have greater deleterious consequences than segmental amplifications. While not necessary, it would be helpful to know the fitness consequences of this deletion in the context of OPC in which AMS3420 or CEC2871 was obtained, a bloodstream model of infection, or a commensal colonization model. In short, why would loss of HGT1 and HGT2 benefit the cell during infection enough to be observed?

First, a point of clarification: the fitness effect of these large CNVs is a combination of the copy number of many genes within the segmental aneuploidy that are either amplified or deleted (e.g. AMS3420 has a whole Chr1 amplification with a 400 kb truncation), and not only a consequence of the ORFs within the repeat sequence. In the Discussion (subsection “Long repeats containing ORFs were significantly more common at breakpoints resulting in CNV, LOH and inversion than any other genomic feature”), we highlight how increased transcription of the HGTs in vivo may cause a DNA DSB that is repaired via HR. The viable outcomes of this recombination event (involving many genes) are what are then selected in vivo.

Breakpoints removed from ALS genes may be due less to poor mapping than rearrangements. How similar was the frequency of called breakpoints in comparable regions encoding tri- or di-nucleotide repeats as are found in the ALS sequences?

To clarify, breakpoints that occurred within the *ALS* gene families were included in Supplementary files 2, 3, and 4 (previously Tables 2, 3, and 4) and comprise one CNV and one LOH breakpoint. For the analysis of inversion breakpoints, the positions that occurred within the *ALS* gene family were removed because the BreakDancer and NUCmer coordinates obtained from Hirakawa et al., 2015 did not support a consistent length of sequence inversion. In these examples, there is likely a structural rearrangement, but high confidence breakpoint positions could not be mapped from these data and were excluded from the final list. The Materials and methods section was modified (subsection “Identification of Inversion Breakpoints”).

Are there features that distinguish between repeat-rich (Chr3R) and repeat-poor (Chr7L) chromosome arms? E.g., GC content, gene density, UTR length, etc., This will be begin to provide predictive correlates to repeats and recombination potential.

Thank you for this question. First, as shown in Figure 2B, the frequency of all long repeat sequences (intra- and inter-chromosomal repeats) correlated with whole chromosome size (excluding the tandem repeat genes, for which we could not accurately determine copy number). The graphical representation of only intra-chromosomal repeat matches (Figure 2A) identified chromosome arms that were repeat-rich or -poor. Therefore, to address this question, we analyzed the repeat density of all repeat matches (intra- and inter-chromosomal) as a function of GC content, ORF density, and chromosome arm size (new Figure 2—source Data 1). No correlation was observed between the number of repeats on a chromosome arm and GC content (linear regression, R^2^ = 0.063, p > 0.32). Similarly, no correlation was observed between the number of repeats on a chromosome arm and the normalized ORF density (ORFs per Mb) (linear regression, R^2^ = 0.02, p > 0.59). A correlation remained between the number of all long repeat sequences and chromosome arm size (linear regression, R^2^ = 0.58, p < 0.0007). In the text we now highlight ChrRR and Chr7L as two examples that deviate most from the linear model (subsection “Identification of long repeat sequences throughout the *C. albicans* genome”, second paragraph).

Reviewer #3:[…] Much is made in both the Results and Discussion section about thelarge 'spacer distance' between intra-chromosomal repeats (e.g. subsection “Identification of long repeat sequences throughout the C. albicans genome”, second paragraph). This is certainly an interesting result, and the raw data makes it clear this is a real phenomenon. However, I think the manuscript could do with somemore clarity about:a) Why this statistic is of interest;b) Precisely what hypothesis is being tested in this "1-way ANOVA withposttest…"I suggest a sentence in the Results section describing the motivation forcalculating this distance. I am not sure what we are mean to glean from the factspacer-distance is not (significantly) correlated with chromosome size (given the small number of chromosomes). It would be good to make the biological motivation for this test explicit or reconsider the test. If the implication is that the repeat-copies are approximately uniformly distributed across chromosomes then a statistical test for this (rather than trend with chromosome size) may be a better test? Alternatively, it may be helpful to simply visualize the distribution of spacer-sizes in each chromosome via a histogram or 1-D kernel estimate).

Thank you for these very helpful suggestions.

The motivation for characterizing the repeat spacer length was added to the Results and Discussion (subsection “Identification of long repeat sequences throughout the *C. albicans* genome”, third paragraph and subsection “*C. albicans* repeat copy length and spacer length”, last two paragraphs). Briefly, the spacer length is important for understanding the origin and evolution of these duplication events in a yeast species that did not undergo an ancient whole genome duplication event. Intra-chromosomal repeats are often generated in tandem by recombination between sister chromatids or replication slippage, and these repeats can move further away from each other by chromosomal rearrangement events (including chromosomal inversions) (Achaz et al., 2000; Reams and Roth, 2015). Therefore, we hypothesized that the repeats would be predominantly tandem, comprised of shorter spacer lengths, and that the distribution of spacer lengths would be the same on each chromosome. The test of these hypotheses found that the median spacer length was very large (~82.8 kb) and that distribution of spacer lengths was significantly different between chromosomes (p < 0.035, Kruskal-Wallis test with Dunn’s multiple comparison).

Previously, the ANOVA with posttest for linearity was used to analyze spacer lengths, normalized for chromosome size, to determine if there was a significant ordered trend (longest chromosome to shortest chromosome) between the spacer lengths located on each chromosome. As the reviewer suggested, when repeat spacer length is plotted according to true chromosome size the two are correlated (new Figure 2—figure supplement 2A, R^2^ = 0.06, p < 0.0001, Figure 2-source data 2). This indicated that repeat spacer length is only limited by chromosome size, however as suggested by the reviewer, this analysis did not provide information about the underlying distribution of spacer length.

To visualize the distribution of spacer lengths on each chromosome we generated new Figure 2—figure supplement 2B and new Figure 2—source data 2. There was a significant difference in the distribution of spacer lengths across all chromosomes (p < 0.035, Kruskal-Wallis test with Dunn’s multiple comparison).

The text has been updated to reflect these new analyses (subsection “Identification of long repeat sequences throughout the *C. albicans* genome”, third paragraph).

I found myself being slightly tripped up by terminology re LTRs andretrotransposons. Presumably, the large number of repeats identified as "LTRS"(e.g. subsection “Identification of long repeat sequences throughout the C. albicans genome”, last paragraph) are long terminal repeats that lack ORFs (non-autonomous or "lone" LTRs) while the small number of retrotransposons will include complete LTR retrotransposons. Perhaps a statement or edit to the Results section making this clear will help readers.

Thank you for this suggestion. We clarified the nomenclature for LTRs and retrotransposons. The two category names were derived directly from the *C. albicans* genome feature file (.gff). Nonetheless, we agreed that additional clarity was needed. We now include “lone” when we introduce LTRs in the Introduction (fourth paragraph), Results (subsection “Identification of long repeat sequences throughout the *C. albicans* genome”, fourth paragraph), and Materials and methods (subsection “Annotation of Repeat Sequences”, first paragraph).